# TeaserGen: Generating Teasers for Long Documentaries

**Weihan Xu**[1]    **Paul Pu Liang**[2]    **Haven Kim**[3]    **Julian McAuley**[3]
**Taylor Berg-Kirkpatrick**[3]    **Hao-Wen Dong**[4]

[1]Duke University    [2]Massachusetts Institute of Technology
[3]University of California San Diego    [4]University of Michigan

weihan.xu@duke.edu, ppliang@mit.edu, {hak004,jmcauley,tberg}@ucsd.edu, hwdong@umich.edu

## ABSTRACT

Teasers are an effective tool for promoting content in entertainment, commercial and educational fields. However, creating an effective teaser for long videos is challenging for it requires long-range multimodal modeling on the input videos, while necessitating maintaining audiovisual alignments, managing scene changes and preserving factual accuracy for the output teasers. Due to the lack of a publicly-available dataset, progress along this research direction has been hindered. In this work, we present *DocumentaryNet*, a collection of 1,269 documentaries paired with their teasers, featuring multimodal data streams of video, speech, music, sound effects and narrations. With DocumentaryNet, we propose a new two-stage system for generating teasers from long documentaries. The proposed *TeaserGen* system first generates the teaser narration from the transcribed narration of the documentary using a pretrained large language model, and then selects the most relevant visual content to accompany the generated narration through language-vision models. For narration-video matching, we explore two approaches: a pretraining-based model using pretrained contrastive language-vision models and a deep sequential model that learns the mapping between the narrations and visuals. Our experimental results show that the pretraining-based approach is more effective at identifying relevant visual content than directly trained deep autoregressive models.

## 1 INTRODUCTION

Teasers are an effective tool for promoting video contents such as documentaries, movies, vlogs, commercials and educational videos. However, creating an effective teaser for long videos possesses unique challenges: first, it requires modeling and understanding long-range multimodal data streams of video, audio and narrations; second, it necessitates maintaining the text-visual correspondence between the teaser narrations and visuals; third, it needs managing smooth scene transitions beyond frame-by-frame audiovisual matching; finally, it entails preserving the factual accuracy in the generated teaser narrations and the accompanying visuals. These challenges together create an exciting yet underexplored ground toward long-range multimodal modeling.

Progress in teaser and trailer generation has been hindered due to the lack of a publicly-available dataset. Existing work on movie trailer generation (Huang et al., 2020; Soldan et al., 2021; Chi et al., 2024) relies on either private datasets or datasets without paired data, creating a barrier for the community to reproduce and follow up their research. In this paper, we present a new documentary dataset with 1,269 high-quality documentaries paired with their teasers. The proposed *DocumentaryNet* dataset features various modalities such as video, speech, music, sound effects, narrations and tags. With the proposed dataset, we explore generating teasers for long documentaries.

In this work, we adopt a narration-centered approach for documentary teaser generation. Given a long documentary, we first generate the teaser narration from the transcribed narration, and then select the most relevant visual content from the documentary to accompany the generated teaser narration. We leverage a pretrained large language model (LLM) with specially-designed prompts to generate teaser narration that has attracting story-like narratives and a thought-provoking ending question. For

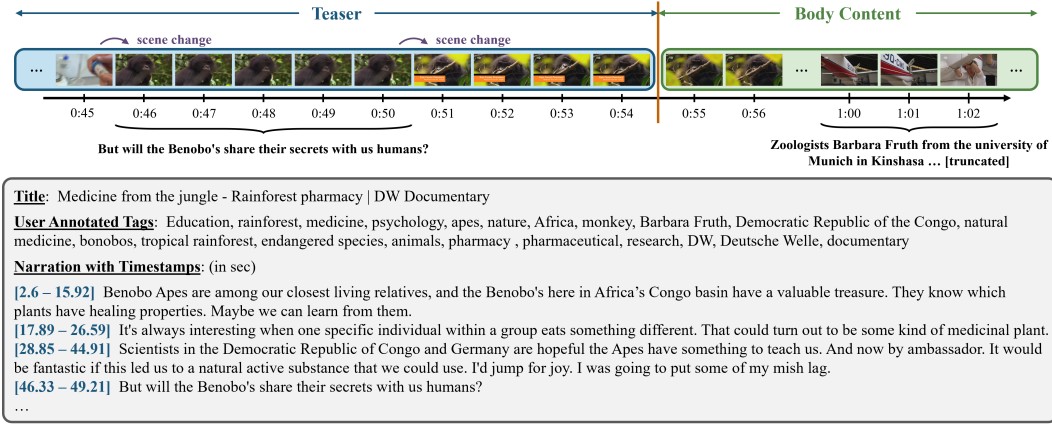

Figure 1: In this paper, we aim to generate teasers, including both the visual and narration, from a long documentary. This figure shows a sample from our *DocumentaryNet* dataset, consisting of 1,269 high-quality documentaries paired with their teasers.

narration-video matching, we first explore framing the task as a constrained optimization problem by applying a thresholding mechanism to maximize the total text-visual matching scores produced by a pretrained language-vision model, while keeping the total length of the selected video clips within a desired range. Further, we explore training a deep sequential model that directly learns the mapping between the narration and the visual content, and we propose various decoding strategies to maximize the narration-visual alignment while maintaining a cohesive output teaser. We compare our propose models with four baseline models (Lin et al., 2023b; Narasimhan et al., 2021b; Son et al., 2024; He et al., 2023a) using objective evaluation metrics and a subjective survey. The experimental results show that the text-visual matching score-based approach is more effective for narration-video matching compared to the directly trained sequential model. Due to the complexity of the task of teaser generation, we focus on narration and visual generation in this paper as an initial step toward the ultimate goal of generating teasers with visually-aligned sound effects and background music. Our contributions can be summarized as follows:

- We propose *DocumentaryNet*, a publicly-available dataset consisting of 1,269 high-quality documentaries paired with their teasers.
- We propose *TeaserGen*, a narration-centered teaser generation system that can effectively compress >30-min documentaries into <3-min teasers.

The DocumentaryNet dataset, along with all source code and demos, can be found on our website. [1]

## 2   RELATED WORK

**Teaser and trailer generation**   Prior work has studied the task of teaser or trailer generation. Chen et al. (2004) proposed to identify and extract semantically important story units and segments of an action movie through movie tempo analysis. Liu & Jiang (2015) learned to select key frames in a semi-supervised manner with a support vector machine. Smeaton et al. (2006) learned to locate the shots that could be contributors to the trailer of an action movie with a support vector machine. However, those three methods do not account for temporal consistency throughout those frames. Argaw et al. (2024a) proposed a unimodal sequence-to-sequence model that selects the key shots from movies without audio outputs. CCANet (Wang et al., 2020) presented a ranking network that uses a co-attention mechanism between movies and trailers to guide the creation of training pairs, with moments strongly correlated with the trailers expected to receive higher scores than those that are less correlated. However, they do not attempt to generate audio narration or text captions to accompany the trailer with temporal correspondence, which is essential in documentary settings. Wang et al. (2024) proposed a human-AI co-creation system to assist video podcasters in crafting

---

[1]https://wx83.github.io/TeaserGen_Official/

Table 1: Comparison of related methods in trailer generation and multimodal summarization

| | Type[*] | Input modality | | | | | Output modality | | | Temporal multimodal information fusion |
|---|---|---|---|---|---|---|---|---|---|---|
| | | Frames | Video | Article | Narration | Query | Text | Video | Frames | |
| MMS (2017) | E | ✓ | ✓ | ✓ | ✓ | ✗ | ✓ | ✗ | ✗ | ✗ |
| CLIP-IT (2021a) | E | ✓ | ✓ | ✗ | ✗ | ✓ | ✗ | ✓ | ✓ | ✗ |
| A2Summ (2023b) | E | ✓ | ✓ | ✗ | ✓ | ✗ | ✓ | ✓ | ✓ | ✓ |
| UniVTG (2023b) | E | ✓ | ✓ | ✗ | ✗ | ✓ | ✗ | ✓ | ✓ | ✗ |
| LfVS (2024b) | E | ✓ | ✓ | ✗ | ✓ | ✗ | ✓ | ✓ | ✓ | ✓ |
| TGT (2024a) | E | ✓ | ✓ | ✗ | ✗ | ✗ | ✗ | ✓ | ✓ | ✗ |
| MSMO (2018) | A | ✓ | ✗ | ✓ | ✗ | ✗ | ✓ | ✗ | ✓ | ✗ |
| MM-AVS (2021) | A | ✓ | ✓ | ✓ | ✓ | ✗ | ✓ | ✓ | ✓ | ✓ |
| VideoXum (2023a) | A | ✓ | ✓ | ✗ | ✗ | ✗ | ✓[†] | ✓ | ✓ | ✓ |
| Ours | A | ✓ | ✓ | ✗ | ✓ | ✓ | ✓ | ✓ | ✓ | ✓ |

[*]**E**–Extractive; **A**–Abstractive    [†] Achieved by dense video captioning

teasers. In this work, we propose a two-stage system that can output narration and visual contents with temporal correspondence.

**Multimodal summarization**    Multimodal summarization aims to create concise summaries by integrating information from multiple modalities. It can be categorized into several types based on how content is extracted and summarized. The first category involves directly extracting text and visual content from the main material. For instance, Argaw et al. (2024b) generates video summaries by leveraging both text and video inputs. They utilize large language models (LLMs) to extract key sentences from transcribed texts, which are then paired with time-aligned video segments to create pseudo-ground truth summaries. A2Summ (He et al., 2023b) produces extractive summaries with a unified multimodal transformer-based model to predict key sentences and its time aligned video segments. Similarly, MMS (Li et al., 2017) generates textual summaries from a set of documents, images, audios, and videos related to a specific topic. Another category of multimodal summarization focuses on obtaining textual information through dense video captioning. For example, Lin et al. (2023a) constructs the VideoXum dataset based on the video captioning dataset ActivityNet (Krishna et al., 2017) and introduces an end-to-end cross-modal video summarization model, VTSUM-BLIP, to achieve video-to-video summarization, video-to-text summarization, and combined video-to-video and text summarization. However, they only input visual contents and generate text summary based on the encoded visual features. A third category involves converting abstractive summaries into extractive labels before identifying key sentences and frames. For instance, MM-AVS (Fu et al., 2021) introduces a Jump-Attention mechanism to align features between text and video by first converting abstractive summaries into extractive labels, then extracting key sentences and frames. In our work, we generate teasers from long documentaries using both narration and visual content. Unlike traditional multimodal summarization, teaser generation requires handling longer input sequences, higher compression rates, and ensuring the narration remains engaging and story-driven.

**Cross-modal alignment**    Recent work in multimodal alignment includes discrete alignment and continuous alignment (Liang et al., 2023). Contrastive learning (Cao et al., 2017; Huang et al., 2017; Grave et al., 2018) is a common approach in discrete local alignment. Optimal transport-based approaches (Villani, 2009) support global alignment between discrete elements across modalities. For continuous alignment, dynamic time warping (Kruskal, 1983) can be used to segment and align multi-view time series data. In this work, we explore using a pretrained contrastive vision-language model as well as directly learning the cross-modal mapping between the narrations and visuals.

# 3    DATASET

We present *DocumentaryNet*, a collection of 1,269 high-quality documentaries from three reputable sources: DW Documentary, Public Broadcasting Service (PBS) and National Geographic. We download the videos and metadata from YouTube, where the metadata include video title, duration, and user-annotated tags. Each documentary in our dataset includes a short teaser at the beginning, which allows us to acquire teaser–documentary pairs through splitting the documentary into its teaser and main content, as shown in Figure 1. We will refer to the main documentary section as the *body content*. To find the boundary between each teaser and its body content, we recruit three annotators

Table 2: Comparison of relevant datasets for video summarization and movie trailer generation. Our dataset is the first publicly available dataset with music and sound effects modality.

| | Modality | | | | Samples | Mean duration (min) | | Compression rate (%) | Publicly available |
|---|---|---|---|---|---|---|---|---|---|
| | Visual | Text | Speech | Music&SFX | | Main content | Teaser | | |
| SumMe (2014) | ✓ | ✗ | ✗ | ✗ | 25 | 2.4 | 0.73 | 30.4 | ✓ |
| TVSum (2015) | ✓ | ✗ | ✗ | ✗ | 50 | 3.9 | 1.17 | 27.9 | ✓ |
| CNN (2021) | ✓ | ✓ | ✓ | ✗ | 203 | 2.1 | 0.12 | 5.7 | ✓ |
| Daily Mail (2021) | ✓ | ✓ | ✓ | ✗ | 1,970 | 1.4 | 0.05 | 3.6 | ✓ |
| BLiSS (2023b) | ✓ | ✓ | ✗ | ✗ | 13,303 | - | 0.17 | - | ✓ |
| MMSum (2024) | ✓ | ✓ | ✗ | ✗ | 5,100 | 14.5 | 0.13 | 0.9 | ✓ |
| LfVS-T (2024b) | ✓ | ✓ | ✗ | ✗ | 1,200 | 12.2 | - | - | ✗ |
| VideoXum (2023a) | ✓ | ✓ | ✗ | ✗ | 14,001 | - | - | 13.6 | ✓ |
| MovieNet (2020) | ✓ | ✓ | ✓ | ✓ | 1,100 | 116 | 1.9 | - | ✗ |
| TGT (2024a) | ✓ | ✓ | ✓ | ✓ | 23,604 | - | - | - | ✗ |
| **DocumentaryNet (ours)** | ✓ | ✓ | ✓ | ✓ | 1,269 | 31.3 | 1.3 | 4.15 | ✓ |

to mark the start of the body content in each documentary. The teasers and the body contents have an average length of 79 seconds and 31.3 minutes, respectively, resulting in a 4.15% compression rate.

To extract narrations from the documentaries, we first separate audio track to three tracks: dialogue, music and sound effects and then use a speech transcription model to transcribe narration from the dialogue track. We adopt a pretrained model (AudioSep, 2023) for sound separation. Since the main content audio is extensive, we split it into 60-second intervals before separating it into the three tracks. To ensure smooth transitions between segments, we apply a windowing function with a window size of 5% based on the standard audio sampling rate of 44,100 Hz. To acquire frame–sentence pairs for training, we adopt Whisper (Whisper-Timestamped, 2023) to transcribe the narration from the dialogues track and estimate the timestamps for each sentence. We also include silence detection information in this dataset for future research. More details of the dataset can be found in Appendix A.

## 4 METHOD

Documentaries often rely on the narration to convey information, while the visual plays a supplementary role in strengthening the narrative. In this paper, we adopt a narration-centered approach by first generating the teaser narration from the transcribed narration of the documentary (Section 4.1) and selecting the most relevant visual content to accompany the teaser narration (Sections 4.2 and 4.2.2).

### 4.1 GENERATING TEASER NARRATIONS BY PROMPTING A LARGE LANGUAGE MODEL

We leverage a pretrained large language model to generate the teaser narration from the transcribed narration of the documentary. We then adopt a text-to-speech model (TTS, 2023) to synthesize the generated script into audio narration. We note that extractive models often fail to construct a coherent narration due to the frequent inserted interviews in documentaries, as exemplified in Appendix B.

To accommodate longer documentaries, we divide the transcribed narration of each main documentary into 10 segments, as the average transcript contains around 3,900 words. For each segment, we prompt GPT-4o (GPT-4o, 2024) to generate a one-sentence summary, resulting in a total of ten single-sentence summaries per documentary. Moreover, since documentary teasers often resemble a story and conclude with a thought-provoking question, we instruct GPT-4o (GPT-4o, 2024) to rewrite the ten summarized sentences into a story-like narration based on the ten summarized sentences[2] and, further, propose an ending question to end the teaser narration.[3] For TeaserGen-PT, as we query the model separately, we retain the names of characters rather than replacing them with pronouns in the story-like narration. We will examine the effectiveness of this approach in Section 5.6.

---

[2]We use the following prompt: "Rewrite the paragraph into an engaging story opening in 10 sentences or less, keeping all names and avoiding being replaced by pronouns."

[3]We use the following prompt: "Given the title and the provided summary, formulate one thought-provoking and concise question that relate directly to the summary."

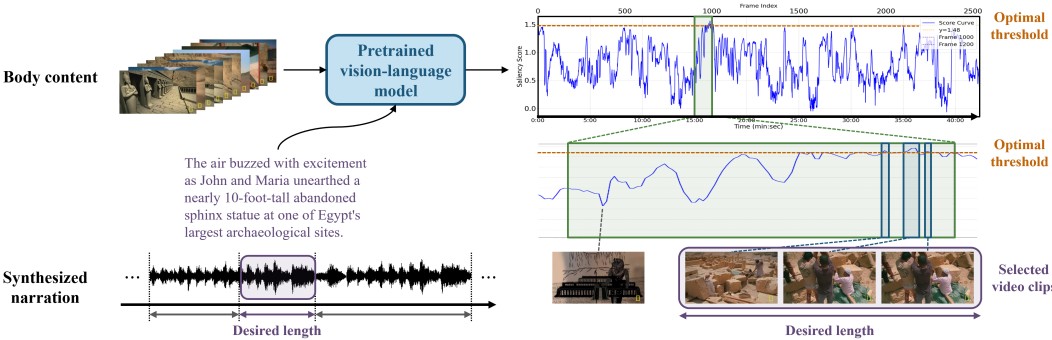

Figure 2: TeaserGen-PT selects video clips from the body content to accompany the generated narration by measuring the matching score between the input sentence and each video frame of the body content and applying a thresholding mechanism to find the optimal video clips. We apply binary search algorithm to find the optimal threshold for each sentence in the narration scripts of the documentaries. We provide the details of the search algorithm in Appendix O.
.

## 4.2 SELECTING ACCOMPANYING VIDEO CLIPS FOR THE TEASER NARRATIONS

To accompany the GPT-summarized narration track with corresponding visual content, we formulate this task as a sentence-by-sentence optimization problem that aims to find the most relevant and important video clips in the body contents to pair with each sentence in the generated narration. Let $\mathcal{V}$ be an input video represented as a sequence of frames sampled at a certain frame rate (1 frame per second in this work), i.e $\mathcal{V} = \{\mathbf{x}_1, \mathbf{x}_2, \mathbf{x}_3, \ldots, \mathbf{x}_n\}$ where $n$ is the total number of frames. Let $\mathcal{S}$ be the input narration represented as a sequence of sentences, i.e., $\mathcal{S} = \{\mathbf{s_1}, \mathbf{s_2}, \cdots, \mathbf{s_m}\}$, where $m$ is the total number of sentences. We first synthesize each sentence $\mathbf{s}_i$ into a waveform, and let $\tau_i$ be the duration of the synthesized speech of sentence $\mathbf{s}_i$. Our goal is to find a number of video clips from the body content that can accompany the generated waveform of sentence $\mathbf{s}_i$ and have a total duration of $\tau_i$. In the following two sections, we will introduce two approaches we propose to tackle this problem.

### 4.2.1 INTERVAL-BASED APPROACH USING PRETRAINED LANGUAGE-VISION MODELS

In the first approach, we frame this task as a constrained optimization problem, where we want to maximize the total text-visual correspondence between each sentence and the selected video clips while making the total length of the selected video clips be the length of the synthesized waveform of the sentence. To achieve this, we use the VTGHL score (VTGHLS) produced by a pretrained video temporal grounding model (Lin et al., 2023b), which measures the importance of the video clips (as defined by whether to be included as a highlight in video highlight detection) as well as the text-visual correspondence. As shown in Figure 2, we first generate a VTGHLS curve for each sentence throughout the main documentary, using a pretrained video temporal grounding model (Lin et al., 2023b). We then adopt binary search to find the threshold value that leads to the desired length. We will refer to this model as *TeaserGen-PT*.

However, we notice this naive implementation leads to overly-fragmented videos with frequent scene changes and repeatedly selected video clips across sentences. To alleviate the overly-fragmented video issue, we only consider clips that are longer than 3 seconds to be selected in our algorithm. Further, to discourage the algorithm from selecting the same video clips across sentences, we only allow one second overlap with each of the video clips selected for the previous two sentences. We note that this issue is prominent in documentaries as a documentary focuses on the same topic and subject throughout the videos.

### 4.2.2 LEARNING-BASED CONTEXT-AWARE APPROACH USING DEEP SEQUENTIAL MODELS

In the second approach, we adopt a train sequential model to learn the mapping between the narration and the visual content. We frame this as a sequence-to-sequence learning where the input is a sequence

Figure 3: An overview of the proposed TeaserGen-LR model——(left) deep sequential model that learns the direct mapping between narration and visual content, and (right) the proposed decoding strategy using beam search and a smoothing mechanism

of sentence embeddings at each frame: $E^{\text{text}} = (\mathbf{e}_1^{\text{text}}, \mathbf{e}_2^{\text{text}}, \cdots \mathbf{e}_n^{\text{text}})$, and the output is a sequence of the corresponding image embeddings: $E^{\text{img}} = (\mathbf{e}_1^{\text{img}}, \mathbf{e}_2^{\text{img}}, \cdots \mathbf{e}_n^{\text{img}})$. In this work, we adopt a transformer model (Vaswani et al., 2023) to learn the mapping $f : E^{\text{text}} \to E^{\text{img}}$. At inference time, we find the frame from the body content that has the closet embedding to the generated image embedding as the selected frame, as shown in Figure 3. We will refer to this model as *TeaserGen-LR*.

In practice, as the transformer operates at the frame level, multiple frames can share the same input sentence embeddings, and tend to generate similar output image embeddings within a sentence, which leads to overly-repeated scenes. To alleviate this issue, we leverage a pretrained diffusion prior model (Ramesh et al., 2022; Prior, 2022) that models the mapping from the CLIP-text embedding space to the CLIP-image embedding space. The diffusion prior model allows the model to generate more diverse output image embeddings, and, further, it helps reduce the modality gap between the CLIP image and text embedding spaces (Liang et al., 2022).

At inference time, we first explore decoding the output image embeddings by finding their nearest neighbors in the body content in the CLIP-image embedding space. In practice, however, we find that this naive approach leads to highly repetitive outputs. This is partly due to the nature of documentaries, which focus on the same topic and subject throughout the video, and thus frames at different parts of the documentaries may be semantically similar and have close CLIP-image embeddings.

To alleviate the overly-repetitive output issue across different sentences, we propose to add an additional regularization term to the decoding algorithm and adopt beam search to find the optimal sequence of frames. Let $\hat{E}^{\text{img}} = (\hat{\mathbf{e}}_1^{\text{img}}, \ldots, \hat{\mathbf{e}}_n^{\text{img}})$ be the generated image embeddings from the model. Our goal is to find a sequence of frames in the body content that have the image embeddings $E^{\star\text{img}} = (\hat{\mathbf{e}}_1^{\text{img}}, \ldots, \hat{\mathbf{e}}_n^{\text{img}})$ that maximize the following score function:

$$\phi(\hat{E}^{\text{img}}, E^{\star\text{img}}) = \sum_{i=1}^n d\left(\hat{\mathbf{e}}_i^{\text{img}}, \mathbf{e}_i^{\star\text{img}}\right) - \lambda \sum_{i,j \text{ in different sentences}} d\left(\mathbf{e}_i^{\star\text{img}}, \mathbf{e}_j^{\star\text{img}}\right), \quad (1)$$

where $d(\cdot, \cdot)$ is the distance metric (cosine similarity in this work) and $\lambda \in \mathbb{R}$ adjusts the strength of the regularization term. The regularization term in Equation (1) encourage the algorithm to find diverse frames from the body content as the match to the raw output image embeddings. We apply beam search with a beam size $K_1$ to find the optimal decoded sequence. For computational efficiency, we consider only the top $K_2$ frames in the body content that are closest to the output embedding as candidates for selection. In this work, we use $K_1 = 5, K_2 = 10$. Finally, to prevent the model from selecting the same frames within one sentence, we also apply a smoothing mechanism that replaces $k$ repeatedly selected frames $\{\mathbf{x}_i, \ldots, \mathbf{x}_i\}$ with nearby consecutive frames centered around $\mathbf{x}_i$: $\{\mathbf{x}_{i-\lfloor \frac{k}{2} \rfloor}, \ldots, \mathbf{x}_{i+k-\lfloor \frac{k}{2} \rfloor -1}\}$.

# 5 EXPERIMENTS AND RESULTS

## 5.1 IMPLEMENTATION DETAILS

Following Argaw et al. (2024a;b), we extract frames from the videos at a rate of one frame per second (1 fps). For TeaserGen-PT, we use CLIP-ViT-B/32 (Radford et al., 2021) for extracting visual and sentence embeddings, chosen to match the embedding size of the pretrained video temporal grounding model. For TeaserGen-LR, we use CLIP-ViT-L/14 (Radford et al., 2021) for embedding extraction,

aligning with the embedding size of the pretrained diffusion prior. The backbone transformer of the diffusion prior consists of 12 blocks with 12 attention heads. For TeaserGen-LR, we utilize 3 transformer layers with a hidden dimension of 768, and we use the L2 distance between the ground truth image embeddings and the generated ones as the loss function. The batch size is set to 16, and we optimize using the Adam optimizer (Kingma & Ba, 2017) with a learning rate of 1e-4. We train the proposed models for 15 epochs and select the best model on the validation set. During evaluation, we track the decoded frame numbers in the body content and use 512-dimensional CLIP embeddings for a fair comparison. We evaluate our model on a test set of 49 documentaries. We include dataset split details in Appendix K. All experiments are conducted on an NVIDIA RTX A6000 GPU.

## 5.2 BASELINE MODELS

We consider six baseline models:[4] (Results for the two video summarization models (He et al., 2023a; Son et al., 2024) can be found in Appendix L.)

- **Random selection**: This baseline model selects clips based on the duration of the generated speech. The model randomly selects $M$ clips from the main content, each corresponding to the duration of the synthesized speech for sentences in $\mathcal{S} = \{\mathbf{s_1}, \mathbf{s_2}, \cdots, \mathbf{s_m}\}$.
- **CLIP-NN**: This baseline CLIP nearest neighbor (CLIP-NN) model selects key frames by finding frames in main documentary that are closet to the input sentences in CLIP embedding space.
- **CLIP-IT** (Narasimhan et al., 2021b): This baseline model generates a teaser by selecting key frames based on any input query. Since CLIP-IT cannot process long video effectively, we divide the original video into 10 sub-clips and pair each sub-clip with a sentence from the GPT-generated script. For each paired sub-clip and sentence, suppose the duration of the synthesized speech of sentence is $\tau_i$, we extract top $\tau_i$ key frames that align with the assigned sentence. Finally, we concatenate all the selected frames in sequence to construct a teaser.
- **UniVTG** (Lin et al., 2023b): This baseline model selects key shots with user-defined keywords or queries to construct a teaser. Since UniVTG cannot take input videos over 10 minutes, we adopt the same approach as in CLIP-IT to construct a teaser.

## 5.3 OBJECTIVE EVALUATION METRICS

To evaluate the performance of our proposed methods, we adopt the following evaluation metrics:[5].

**Retrieval-based metrics**    Following Hesham et al. (2018); Argaw et al. (2024a), we calculate the F1 score by comparing the frames selected in the generated teaser to those of the ground truth. While the F1 score provides us an estimate on the overlap between generated teaser and the ground truth, they should not be treated as the only standards as teaser generation is inherently a generative task.[5]

**Repetitiveness**    To measure the repetitiveness of generated teaser, we propose a new metric:

$$\text{REP} = 1 - \frac{\text{Number of unique frames}}{\text{Number of frames}} \tag{2}$$

The repetitiveness score computed on the ground truth is 7.86%, suggesting that there are some repeated frames in the ground truth teasers.[5]

**Scene change rate (SCR)**    To measure the degree of fragmentation, we propose the SCR metric:

$$\text{SCR} = 1 - \frac{\text{Number of consecutive frames within the same scene}}{\text{Number of frames}} \tag{3}$$

SCR measures the frequency of scene changes of the teasers. The estimated SCR of the ground truth teasers is 27.6% by manual inspections on 10 video samples.[5]

**CLIPScore**    To evaluate the alignment between the narration and the visual content, we compute the CLIPScore (Hessel et al., 2021). Let $\mathbf{e}_k^{\text{img}}$ and $\mathbf{e}_k^{\text{text}}$ be the text and image embeddings at frame $k$, and $\text{cos\_sim}(\cdot)$ be the cosine similarity. CLIPScore is defined as

$$\text{CLIPScore} = 2.5 \cdot \frac{1}{n} \cdot \sum_k^n \max\left(\text{cos\_sim}\left(\mathbf{e}_k^{\text{img}}, \mathbf{e}_k^{\text{text}}\right), 0\right), \tag{4}$$

---

[4]We intended to compare our model to Argaw et al. (2024a). However, we cannot reproduce their results as it is trained on a private dataset, and their code and pretrained models are not available.

[5]Due to space concern, we provide details of the calculation in Appendix C . We also include an objective evaluation on generated narration scripts in Appendix I

Table 3: Objective evaluation results with LLM-generated narrations.

| Model | Query | Decoding | DP | F1 (%)↑ | REP (%) | SCR (%) | CLIPScore | VTGHLS |
|---|---|---|---|---|---|---|---|---|
| **Baseline models** | | | | | | | | |
| Random | Random | - | - | 1.67 | 4.05 | 7.81 | 0.56 | 0.75 |
| CLIP-NN | Narration | Greedy | × | 0.11 | 92.73 | 8.29 | 0.69 | 0.79 |
| UniVTG (2023b) | Title | Rank | - | 1.82 | 0 | 89.68 | 0.58 | 1.01 |
| CLIP-it (2021b) | Narration | Rank | × | 1.24 | 0 | 99.39 | 0.56 | 0.61 |
| **Pretraining-based models** | | | | | | | | |
| TeaserGen-PT | Title | Thresholding | - | 1.85 | 0 | 13.16 | 0.56 | 1.02 |
| TeaserGen-PT | Narration | Thresholding | - | 1.07 | 21.38 | 22.58 | 0.58 | 1.45 |
| TeaserGen-PT-CLIP | Narration | Threshold | × | 1.31 | 27.23 | 24.10 | 0.58 | 0.74 |
| **Learning-based models** | | | | | | | | |
| TeaserGen-LR | Narration | Greedy | × | 1.56 | 31.97 | 27.18 | 0.58 | 0.74 |
| TeaserGen-LR | Narration | Greedy | ✓ | 1.38 | 26.83 | 35.48 | 0.62 | 0.78 |
| TeaserGen-LR | Narration | Beam search | × | **1.88** | 24.16 | 41.97 | 0.58 | 0.74 |
| TeaserGen-LR | Narration | Beam Search | ✓ | **1.88** | 19.39 | 46.56 | 0.63 | 0.77 |
| Ground truth | - | - | - | 100 | >7.86 | 27.6 | 0.58 | 0.64 |

**VTGHLS**   Proposed by Lin et al. (2023b), VTGHLS measures the importance of the frame (as defined by whether to be included as a highlight in video highlight detection) in addition to the text-visual correspondence. We estimate a VTGHLS of 0.64 for the ground truth teasers.[5]

## 5.4   OBJECTIVE EVALUATION RESULTS

As shown in Table 3, the proposed TeaserGen-PT model that uses the title as the query outperforms both baseline models as indicated by the higher F1 score and a closer scene change rate to that of the ground truth. Similarly, TeaserGen-LR, when using beam search, achieves a higher F1 score than than the baseline models and a closer scene change rate to that of the ground truth. In contrast, the two baseline models, i.e., UniVTG (Lin et al., 2023b) and CLIP-it (Narasimhan et al., 2021b), both lead to a scene change rate that is three to four times greater than that of the ground truth. Further, while the CLIP-NN baseline achieves the highest CLIPScore, it results in high repetitiveness, partly because the nearest neighbor search can easily lead to repeatedly selected video clips. Moreover, TeaserGen-LR equipped with the diffusion prior model achieves us the second highest CLIPScore, demonstrating the effectiveness of diffusion prior in bridging the gap between the CLIP image and text embedding spaces. We also observe that transformer-based models (TeaserGen-LR) tends to result in a higher scene change rate than the pretraining-based models (TeaserGen-PT).

For TeaserGen-PT models, while it achieves a higher CLIPScore when using narration as the query, it achieves a higher F1 score when using title as the query. This is possibly because a documentary often revolves around a specific topic or character, which is usually captured by the title, whereas narrations provide more detailed but denser descriptions. For TeaserGen-LR models, in terms of F1 score, TeaserGen-LR, with beam search and diffusion prior, outperforms other transformer-based models. We find that decoding with the beam search method proposed in Section 4.2.2 can effectively reduce repetitiveness by discouraging the selection of clips from different scenes.

## 5.5   SUBJECTIVE EVALUATION

To further measure the quality of our generated scripts and teasers, we conduct a subjective test. We randomly select 10 documentaries from the test dataset and split them into 2 versions, with each version containing demos for 5 documentaries. We then assess the coherence, video-narration alignment, engagingness, and realness (see Appendix D for the questions we ask in the survey) of the generated teasers on a Likert scale from 1 to 5. Additionally, we ask participants to compare the effectiveness of two approaches: summarizing each chunk of content directly versus using our carefully designed GPT prompts, also on a Likert scale from 1 to 5. This comparison focuses on how both methods perform in terms of consistency, informativeness, and engagingness. We recruit 20 participants for the subjective test, with 11 working on version A and 9 working on version B. Among all participants, 14 out of 20 have video editing experience.

We report in Table 4, the mean values from subjective tests, along with the 95% confidence intervals. Human evaluations further confirm that TeaserGen-PT surpasses baseline models with higher scores

Table 4: Subjective evaluation result with LLM-generated narrations.

| Model | Query | Decoding | Coherence↑ | Alignment↑ | Engagingness↑ | Realness↑ |
|---|---|---|---|---|---|---|
| UniVTG (2023b) | Title | Rank | $2.61 \pm 0.50$ | $2.62 \pm 0.47$ | $2.67 \pm 0.57$ | $2.66 \pm 0.54$ |
| CLIP-it (2021b) | Narration | Rank | $2.61 \pm 0.46$ | $2.67 \pm 0.44$ | $2.57 \pm 0.46$ | $2.51 \pm 0.46$ |
| TeaserGen-PT | Title | Threshold | $\mathbf{3.14 \pm 0.50}$ | $2.84 \pm 0.57$ | $\mathbf{2.81 \pm 0.49}$ | $\mathbf{2.94 \pm 0.50}$ |
| TeaserGen-LR | Narration | Greedy | $2.90 \pm 0.45$ | $\mathbf{2.88 \pm 0.48}$ | $2.71 \pm 0.42$ | $2.71 \pm 0.44$ |
| TeaserGen-LR | Narration | Beam search | $2.84 \pm 0.46$ | $2.69 \pm 0.51$ | $2.71 \pm 0.42$ | $2.64 \pm 0.41$ |

Table 5: Subjective evaluation results of the LLM-generated narrations

| Narration | Organization↑ | Informativeness↑ | Engagingness↑ |
|---|---|---|---|
| Naive summarization | $3.58 \pm 0.57$ | $3.72 \pm 0.47$ | $3.60 \pm 0.56$ |
| Finely-tuned scripts | $\mathbf{3.88 \pm 0.44}$ | $\mathbf{3.82 \pm 0.54}$ | $\mathbf{3.70 \pm 0.46}$ |

in terms of coherence, alignment, engagingness, and realness. The low coherence of baseline model in human study further proves that the baseline model is overly-fragmented. While TeaserGen-LR with beam search decoding has a higher F1 score and CLIPScore compared to TeaserGen-PT using the title as the query in objective metrics, subjective evaluations reveal that it performs lower in terms of coherence, alignment, engagingness, and realness. This discrepancy is likely due to its higher scene change rate, which reflects more fragmented sequences and quicker transitions between clips. We encourage the readers to watch the video samples on our demo website.[1]

## 5.6 ABLATION STUDY

**Experiment on changing the matching score function**   In this experiment, we compare using CLIPScore (Radford et al., 2021) versus VTGHLS (Lin et al., 2023b) as the matching metric in Section 4.2.1 to examine the effects of changing the matching score function. Unlike CLIPScore, VTGHLS consider both the importance of a frame and the text-visual correspondence. The goal of this experiment is to examine whether incorporating the importance of a frame is beneficial for video-narration matching. As shown in Table 3, we find that the model that uses VTGHLS as the matching score function outperforms that uses CLIPScore instead, resulting in higher F1 score, lower repetitiveness and lower scene change rate, closer to that of the ground truth. This highlights the benefits of using a matching score that takes into account the importance of a frame.

**Effectiveness of the proposed prompting approach for teaser narration generation**   In this experiment, we examine the teaser generation approach proposed in Section 4.1. We conduct a subjective listening test to compare the teaser narrations generated by a naive summarization prompt and those generated by our proposed prompting approach in terms of organization, informativeness and engagingness (see Appendix E for the questions we ask in the survey). We report in Table 5, the mean values from subjective tests, along with the 95% confidence intervals. The results indicate that our finely-tuned prompts outperform naive summarization methods across all three dimensions. This indicates story-like conversion and ending questions can make narrations scripts closer to human expectations. We provide in Appendix B examples of the extractive summary, ground truth narration, LLM with non-finetuned prompts and LLM with finetuned prompts.

**Effects of the smoothing mechanism and the diffusion prior model**   In this experiment, we examine the effectiveness of the smoothing mechanism and the diffusion prior on the teaser generation approach proposed in Section 4.2.2. Table 6 shows that applying smoothing mechanism can significantly decrease the repetitiveness as well as increase the F1 score, demonstrating the effectiveness of the smoothing mechanism proposed in Section 4.2.2. Further, Table 6 shows that applying diffusion prior leads to lower repetitiveness, higher language-vision correspondence (as indicated by higher CLIP and VTGHLS). The decreased repetitiveness possibly result from the sampling process in diffusion prior, which is a generative model that generate different image embeddings for the same input text embeddings. The increased CLIPScore possibly come from the fact that diffusion prior can bridge the gap between textual embeddings and visual embeddings. Comparing the two results of TeaserGen-LR using beam search decoding, we also find applying diffusion prior can result in a higher scene change rate, partially because diffusion prior would further encourage scene changes.

In addition to the above ablation study, we present additional experimental results exploring the impact of $\lambda$, inference using ground truth narration, and the effects of minimum clip length and

Table 6: Effects of the smoothing mechanism and diffusion prior for TeaserGen-LR

| Decoding | DP | Smoothing | F1 (%)↑ | REP (%) | SCR (%) | CLIPScore | VTGHLS |
|---|---|---|---|---|---|---|---|
| Greedy | ✗ | ✗ | 0.53 | 81.32 | 27.38 | 0.59 | 0.74 |
| Greedy | ✗ | ✓ | 1.56 | 31.97 | 27.18 | 0.58 | 0.74 |
| Greedy | ✓ | ✓ | 1.38 | 26.83 | 35.47 | 0.62 | 0.78 |
| Beam search | ✗ | ✓ | **1.88** | 24.16 | 41.97 | 0.58 | 0.74 |
| Beam Search | ✓ | ✓ | **1.88** | 19.39 | 46.56 | 0.63 | 0.77 |

overlapping tolerance in TeaserGen-PT, as detailed in Appendices F, H and N. Qualitative examples are also available on our demo page.

## 6 Discussion and Limitations

**Alignment between objective and subjective evaluation results**  From Tables 3 and 4, we find that the scene change rate metric aligns well with the coherence and engagingness scores in our subjective evaluation, where a lower scene change rate is preferable. Several survey participants also pointed out that too frequent scene changes negatively affect the viewer experience. Further, we notice that higher CLIPScore does not always lead to a higher alignment score in human evaluation. This is possibly because CLIPScore is calculated on the frame-level, while humans perceive narration-video correspondence on the clip-level. This necessitates a clip-level narration-video correspondence metric for objective evaluation in future work, which is not trivial as it inherently involves scene change detection and clip-level video understanding.

**Limitations and future work**  Finally, we want to point several notable limitations of our work. First, our proposed model leverage several assumptions that might not hold for other media. For example, we assume that a scene change always happens when we move from one sentence to next in the narration. In addition, the our proposed two-stage approach inherently assume that narration plays a more significant role than visual content, which is a reasonable assumption for many documentaries but might not work for more visual-centered media such as vlogs and silent movies. Second, the proposed narration-video matching approach cannot accurately match interview scenes commonly seen in documentaries as the pretrained language-vision model cannot associate names with their corresponding faces, which would require a separate video understanding module. Third, teaser narration generation is a creative process. While we have proposed several prompting strategies for generating better teaser narrations, the proposed method still falls short in terms of artistic quality and creativeness compared to scripts created by professionals. Last but not least, we only consider documentary teaser generation in this paper due to the scarcity of public datasets for other media. However, to examine the generalizability of our proposed models, we apply our proposed models to other media in a zero-shot setting, including old movies and educational videos. We provide some qualitative examples of the generated teasers on the demo website [1] and objective evaluations on the teasers created in a zero-shot scenario in Appendix M. For future work, we would like to further scale up the dataset and further explore end-to-end approaches for teaser generation. Moreover, we would also like to explore generating music and sound effects to accompany the generated teasers.

## 7 Conclusion

We have presented a new documentary dataset, DocumentaryNet, that consists of 1,269 documentary-teaser pairs with multimodal data streams of video, speech, music, sound effects, narrations and tags. With DocumentaryNet, we have proposed the TeaserGen system for generating teasers for long documentaries. We adopt a narration-centered approach and approach teaser generation by first generating the narration using a pretrained LLM and then selecting accompanying video clips from the main content. We have explored two approaches for narration-video matching: TeaserGen-PT is based on a pretrained contrastive language-vision model and a thresholding mechanism, whereas TeaserGen-LR is a learning-based model that directly models the mapping between the narrations and visuals. Through objective and subjective evaluation, we have demonstrated the effectiveness of the proposed system against several baseline models. We hope our work pave a pathway towards long-range multimodal modeling by exploring this new task of documentary teaser generation.

## ETHICS STATEMENT

Our proposed models do not create new narrations and visuals from scratch; instead, they focus on summarizing, reorganizing, and rewriting existing content. In the narration generation phase, our approach does not involve training the model with copyrighted materials. We utilize GPT-4 to generate the teaser narration by summarizing and rewriting the original narration. In the visual matching phase, for TeaserGen-PT, we employ a contrastive language-vision model to select video clips from the main documentary, rather than deploying a generative model. For TeaserGen-LR, we have developed a generative model that generates CLIP-image embeddings from the narrations. We note that TeaserGen-LR inherently learn the underlying distributions of DocumentaryNet in the CLIP embedding space, but these CLIP-image embeddings need to be further decoded into video frames. In this work, we use a nearest neighbor based algorithm to decode the generated CLIP-image embeddings by finding the closest video frames in the original documentary, which does not involve learning the underlying distribution of DocumentaryNet.

Moreover, we note that our system may produce inaccurate teasers where faces are paired with the wrong names or where the generated narrations are paired with irrelevant visuals. Further investigation is needed to avoid unintended misinformation or hallucination of the proposed system. Despite these challenges, we envision practical and significant applications of our model such as generating educational content previews and enhancing video accessibility. We hope our work contribute to the long-term goal of enriching human experience and augmenting human creativity through thoughtful use of AI technology. For copyright concerns, we will not distribute the raw videos from our datasets, but we will provide access to publicly available YouTube links and data preprocessing scripts.

## REPRODUCIBILITY STATEMENT

For reproducibility, we will release all the source code. Due to copyright concerns, we will release only the YouTube links to the videos, along with the metadata and annotations.

## ACKNOWLEDGEMENT

We would like to thank Ruoyu Wang and Hongyi Yin for helping with the data annotation process. This work was partially supported by the NSF under grants 2146151 and 2200333.

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

## A   DETAILS OF THE DOCUMENTARYNET DATASET

**Three-stream Sound Separation**   We extract the audio track of the documentary using the ffmpeg library. We then use a pretrained three-stream audio separation model (AudioSep, 2023) to separate each audio track into three stems: *dialogue*, *music* and *sound effects*. Due to the length of the audio track of the documentary, we split each documentary into 60-second chunks and then run the separation model chunk by chunk. Finally, we smooth the transitions with a window size equal to 5% of the audio sampling rate (44,100 Hz in this work). While we do not use the music and sound effect tracks in this work, we include them in the DocumentaryNet dataset as we believe they would be helpful for future research on teaser generation.

**Silence Detection**   we use a silence detector (Slicer, 2022) to detect periods of silence within each separated audio track. We randomly select 20 samples to select the optimal threshold value for classifying segments as noise or sound. A binary label is assigned to each segment, with '1' indicating sound and '0' indicating silence. The final thresholds used are -40 dB for the music track, -25 dB for the dialogue track, and -40 dB for the sound effects track.

## B   COMPARISON OF THE GENERATED NARRATIONS

We compare original narration, LLM-generated teaser narrations, LLM-generated teaser narrations with finetuned prompts in Tables 11 and 12. We leverage T5 (Text-to-Text Transfer Transformer) model to generate extractive summary on transcribed narration.

## C   DETAILS OF THE EVALUATION METRICS

**Ground truth matching**   In this work, we only consider those non-fully black video clips that can be extracted from body contents. Therefore, we remove those without match. In order to find those without match, we randomly pick 15 documentaries with teaser and body contents frame pairs. Considering the low FPS, there might be some shift when we extract frames. Therefore, due to frame shift, we allow a higher tolerance in distance. We calculate the lowest 20 % $L_2$ distances between each teaser frame and each body contents frame. The selected lowest distance threshold is 88.92. To find the matched interval, we first extract CLIP feature with CLIP(L-14). We select top 20

images with the highest cosine similarity to find those close in semantic meanings. Then we calculate pixel-by-pixel L2 distance between teaser frame and those 20 images. We pick the one with the lowest distance and smaller than the selected threshold. In order to remove dark frames, we randomly select frames from the body content of 10 documentaries and calculate the average brightness of their raw pixels. We define the darkest frames as those in the lowest 5% of brightness.

**Repetitiveness**   Since we are not able to find some of the frame in body contents due to frame shift, we calculate the lower bound of repetitiveness by considering all non matched frames as different frames. The lower bound is 0.0786.

**Scene Change Rate**   The upper bound is calculated by considering all consecutive non matched frames as a new clip as those consecutive non matched frames might contain more than one clip in practice. We also randomly pick 10 documentaries teaser and calculate number of video clips per teaser. The average number of clips per teaser is around 27.6 %.

**VTGHLS**   For each sentence in the ground truth teaser, we first generate a score function using Lin et al. (2023b). This function measures frame importance (defined by its likelihood of being included as a highlight in video highlight detection) along with text-visual correspondence. Next, we calculate the VTGHLS for each matched frame. By averaging VTGHLS values over all matched frames of all documentaries in DocumentaryNet, we obtain an estimated VTGHLS of 0.64.

The estimated VTGHLS value of 0.64 for the ground truth, as reported in Table 3, is included solely for comparison purposes. It is not used at any stage in our training or inference pipeline.

# D   SUBJECTIVE EVALUATION QUESTIONS FOR TEASER GENERATION

We ask the following questions in our subjective evaluation survey to evaluate our proposed models for teaser generation, as described in Section 5.5 and reported in Table 4.

- **Coherence**: "To what extent do you feel that the sample maintains coherence and a smooth flow, ensuring that each segment transitions logically and the overall experience feels seamless?"
- **Correspondence**: "To what extend do you feel that the narration and a video match and work well together, making the overall presentation clear and easy to follow?"
- **Engagingness**: "To what extent do you feel that the sample captures your interest and keep you engaged throughout?"
- **Realness**: "How well do you feel this sample meets your expectations as a teaser for a documentary in general?"

# E   SUBJECTIVE EVALUATION QUESTIONS FOR TEASER NARRATION GENERATION

We ask the following questions in our subjective evaluation survey to evaluate our proposed approach for generating teaser narration, as described in Section 4.1 and reported in Table 5.

- **Organization**: To what extent do you feel that the narration present a well-organized and consistent storyline?
- **Informativeness**: To what extent do you feel that the narration introduces the main characters and conflicts clearly and concisely?
- **Engagingness**: To what extend do you feel the narration keeps you engaged and builds curiosity about what happens next?

# F   EXPERIMENT ON INFERENCE WITH GROUND TRUTH NARRATIONS

Although we cannot have ground truth narration when making teasers in reality, we conduct an ablation study to compare model behavior on real narrations and machine generated narrations. We did not apply the diffusion prior model for TeaserGen-LR in this experiment. We report objective evaluation results in Table 7 and subjective evaluation results in Table 8. As shown in Table 8,

Table 7: Objective evaluation results for the experiment on inference with ground truth narrations

| Backbone | Query | Decoding | Smoothing | F1 (%)↑ | REP (%) | SCR (%) | CLIPScore | VTGHLS |
|---|---|---|---|---|---|---|---|---|
| **Baseline models** | | | | | | | | |
| Random | Random | - | - | 1.65 | 1.60 | 12.92 | 0.54 | 0.64 |
| UniVTG (2023b) | Title | Rank | - | 1.67 | 0 | 88.50 | 0.55 | 0.79 |
| CLIP-it (2021b) | Narration | Rank | - | 1.20 | 0 | 99.24 | 0.54 | 0.44 |
| **Pretraining-based models** | | | | | | | | |
| TeaserGen-PT | Title | Threshold | - | 1.46 | 0 | 17.33 | 0.55 | 1.03 |
| TeaserGen-PT | Narration | Threshold | - | 1.20 | 6.12 | 24.42 | 0.55 | 1.40 |
| TeaserGen-PT-CLIP | Narration | Threshold | - | 1.90 | 7.79 | 23.85 | 0.56 | 0.72 |
| **Learning-based models** | | | | | | | | |
| TeaserGen-LR | Narration | Greedy | × | 0.88 | 68.87 | 34.19 | 0.57 | 0.71 |
| TeaserGen-LR | Narration | Greedy | ✓ | 1.76 | 11.91 | 34.07 | 0.56 | 0.71 |
| TeaserGen-LR | Narration | Beam search | ✓ | 2.20 | 10.16 | 45.84 | 0.56 | 0.72 |
| Ground truth | - | - | - | 100 | >7.86 | 27.6 | 0.58 | 0.64 |

Table 8: Subjective evaluation results for the experiment on inference with ground truth narrations

| Model | Query | Decoding | Coherence↑ | Correspondence↑ | Engagingness↑ | Realness↑ |
|---|---|---|---|---|---|---|
| UniVTG (2023b) | Title | Rank | $2.64 \pm 0.53$ | $2.72 \pm 0.60$ | $2.76 \pm 0.53$ | $2.64 \pm 0.57$ |
| TeaserGen-PT | Title | Threshold | $\mathbf{3.45 \pm 0.47}$ | $\mathbf{2.96 \pm 0.66}$ | $\mathbf{2.91 \pm 0.58}$ | $\mathbf{2.97 \pm 0.63}$ |
| TeaserGen-LR | Narration | NN | $2.93 \pm 0.58$ | $2.69 \pm 0.62$ | $2.74 \pm 0.62$ | $2.61 \pm 0.67$ |

TeaserGen-PT achieves the highest score in terms of coherence, correspondence and engagingness in human evaluation.

# G  EXPERIMENT ON FINETUNING THE PRETRAINED VIDEO TEMPORAL GROUNDING MODEL

We prepare highlight detection dataset by constructing query and highlight frame pairs with timestamped narration. We have around 10,333 samples to finetune the pretrained UniVTG. However, Table 9 shows the finetuned model result in a worse performance due to the relatively small size of the finetuning dataset comparing with the dataset in pretraining.

# H  EFFECTS OF THE REGULARIZATION TERM $\lambda$

We show in Table 10 the results when using different values of $\lambda$ in Equation (1). As shown in Table 10, we find that a higher $\lambda$ can reduce the repetitiveness and lead to a higher F1 score but the difference is not significant.

Table 9: Objective evaluation results for the finetuning experiment with TeaserGen-PT

| Query | Decoding | Finetune | F1 (%)↑ | REP (%) | SCR (%) | CLIPScore | VTGHLS |
|---|---|---|---|---|---|---|---|
| Title | Thresholding | × | 1.85 | 0 | 13.16 | 0.56 | 1.02 |
| Narration | Thresholding | × | 1.07 | 21.38 | 22.58 | 0.58 | 1.45 |
| Narration | Thresholding | ✓ | 1.23 | 41.39 | 24.01 | 0.56 | 1.05 |

Table 10: Effect of the regularization term $\lambda$ in Equation (1)

| Decoding | $\lambda$ | F1 (%)↑ | REP | SCR | CLIPScore | VTGHLS |
|---|---|---|---|---|---|---|
| Beam search | 1 | 1.88 | 24.16 | 41.97 | 0.58 | 0.74 |
| Beam search | 10 | 1.91 | 24.01 | 41.78 | 0.58 | 0.74 |
| Beam search | 100 | 1.91 | 24.06 | 41.73 | 0.58 | 0.74 |

## I  OBJECTIVE EVALUATION ON LLM-GENERATED NARRATION

To assess the quality of generated narration scripts, we report common text summarization evaluation metrics: F1 of ROUGE in Table 13 by comparing our LLM-Generated narration with audio transcription of ground truth teaser. The referred text is from the audio transcription of ground truth teaser where dialogues are also contained. We also compare the TeaserGen performance with fully extraction from body contents narration and naive summarization. Note that our generated narrations intentionally exclude dialogue and include an engaging ending question, which inherently reduces overlap and leads to lower ROUGE scores. To address this limitation, we also conduct subjective evaluation metrics to better assess the quality of our generated narrations, as ROUGE alone does not fully capture the nuances of narration quality in this context.

Table 13: Objective evaluation results of the LLM-generated narrations

| Evaluation | Naive Summarization | Fully Extraction | Ours |
|---|---|---|---|
| ROUGE-1 | 0.14 | 0.15 | 0.14 |
| ROUGE-2 | 0.02 | 0.02 | 0.02 |
| ROUGE-l | 0.12 | 0.14 | 0.12 |

## J  COMPARISON WITH EXISTING TEASER/TRAILER-BODY CONTENTS DATASET

In Table 14, we compare DocumentaryNet with two existing teaser/trailer-body contents dataset. DocumentaryNet is the first publicly available teaser/trailer-body contents dataset.

Table 14: Comparison with Existing Teaser/Trailer-Body Contents Dataset

| Dataset | Number of Samples | Total Lengths(hours) | Public Available |
|---|---|---|---|
| MovieNet (2020) | 1100 | 2126.7 | ✗ |
| TGT (2024a) | 22,604 | - | ✗ |
| Ours | 1,269 | 655.7 | ✓ |

## K  DATASET SPLIT DETAILS

In Table 15, we show the details of dataset splitting.

Table 15: Dataset Split Details

| Split | Number of Samples | Total Hours |
|---|---|---|
| Train | 1026 | 514 |
| Validation | 57 | 32 |
| Test | 49 | 29 |

Table 11: Comparison of LLM-generated teaser narrations—Example I

| Model | Narration |
|---|---|
| Extractive | archaeologists are searching for clues to reveal more about the pharaoh queen. hetshepsood's temple is the key to the secrets of her dynasty, says dr zbigniew szybranski, a doctor of ranski team. the temple was buried and badly damaged and little was known about its owner. a ram-headed sphinx has been excavated at one of the largest archaeological sites in ancient Egypt. it's thought that Hedge Sheppset started the greatest display of Sphinces known to ancient Egyptians. the necropolis of around 100 ancient tombs is 120 miles from Luxor in Aswan. cnn's richard quest is excavating what could be one of the sphinxes of Hatshepsut. quest finds a small cryo-like body in the quarry at gopel, in southern egypt, near the tomb of king henry ibn al-qaeda - he was buried there in 1586 b.c. doctors of ransky is investigating the paintings she left on the walls of head chef's at temple. the imagery holds clues to her life and reveals a family power struggle with her nephew, who is also her stepson, Tutmost III. despite being female, Hadshepsut is often depicted as a man. but hidden in the hieroglyphic text, Colleen finds evidence of her real gender. at the Necropolis in Aswan, a team finds an ancient face mask made of cartonage. it was skillfully how they made the eyes of an adult mummy buried outside the tomb. the mask was meant to help the afterlife from gold, this one is more beautiful than to come once again. american archaeologist finds dramatic evidence of one individual's life and death. the vagina is the whole here, still very distended, so we know within 24 hours of giving birth, that she died. but the female pharaoh had chefsuit wanted to be more than just equal. doctors of Ransky are working to restore temple of Hachepsut. ancient earthquakes and landslides have damaged the temple's upper terrace, which is 11 feet higher than the valley of the kings. the bones and delicate cartonage have been moved into a lab in aswan, savan. hundreds of sphinxes still remain at carnax temple site. a ten-foot high abandoned statue is the largest they've seen at the quarry, john and maria Ward say. the couple are looking for black inclusions within the stone itself. there are no records whatsoever of any unfinished statue that is intact like this. sphinxes from sosilla were shipped 100 miles down the nile to carnac temple. the evidence suggests they came from the city, not so far from where they are now - john w. mcdonald jr. |
| LLM-generated | Archaeologists, led by Dr. Zbigniew Szybranski, are excavating and restoring the unique temple of the revolutionary female pharaoh Hetshepsood in Luxor, Egypt, to uncover more about her enigmatic reign and contributions, while other teams investigate her quarries to understand her extensive building projects John and Maria are excavating an abandoned statue at one of Egypt's largest archaeological sites, while Martina Bartanova's team in Aswan discovers ancient human remains that could halt their exploration of an unopened tomb. Martina and her team are excavating a site where they discover a child's remains and a miniature sphinx, while dealing with the challenges of preserving delicate artifacts and documenting their findings for the Egyptian government. The paragraph describes various archaeological efforts in Egypt, including the discovery of a miniature model likely carved for practice, the investigation of Queen Hatshepsut's rise to power and her conflict with her stepson, the challenges faced by a team uncovering artifacts near her temple, and the work of Yale professors using digital technology to study ancient inscriptions. The couple explores Karnak Temple to investigate inscriptions revealing Hadshepsut's portrayal as a male pharaoh, while archaeologists at Dra'abu El Naga study the roles of women in ancient Egyptian society through the excavation of a tomb filled with fragmented human remains. The team is meticulously organizing human remains, including mummified organs and detached body parts, while making significant discoveries such as a young woman who likely died in childbirth and an ancient face mask, all under the pressure of impending strong winds. American archaeologist Susanne Onstein is uncovering the roles and mortal dangers faced by women in ancient society, while also exploring the monumental building campaigns of female pharaoh Hatshepsut, who used colossal obelisks as both religious devotion and propaganda to immortalize her reign. The excavation team has unearthed ancient pottery fragments and evidence of temples built by Hatshepsut and her stepson Tutmose III, revealing a historical power play and ongoing restoration efforts, while another team in Aswan uncovers Middle Kingdom burials, all amidst the personal reflections and challenges faced by the archaeologists. John and Maria unearthed a nearly 10-foot-tall abandoned sphinx statue at a quarry and are now investigating at the Carnax temple to find a matching sphinx, focusing on identifying black inclusions in the sandstone to determine its origin. John and Maria's discovery of sphinxes from Sosilla, transported to Carnac Temple, alongside Dr. Ransky's excavation and restoration efforts, has unearthed significant artifacts and revived the legacy of the remarkable Pharaoh Queen Hatshepsut. |
| LLM-generated (tuned) | Under the scorching Egyptian sun, Dr. Zbigniew Szybranski led a team of archaeologists in Luxor, meticulously excavating the unique temple of the revolutionary female pharaoh Hetshepsood. The air buzzed with excitement as John and Maria unearthed a nearly 10-foot-tall abandoned sphinx statue at one of Egypt's largest archaeological sites. Meanwhile, in Aswan, Martina Bartanova's team stumbled upon ancient human remains that could halt their exploration of an unopened tomb. The discovery of a child's remains and a miniature sphinx added layers of mystery to their quest. At Karnak Temple, John and Maria delved into inscriptions revealing Hetshepsood's portrayal as a male pharaoh, while Yale professors used digital technology to study ancient texts. In Dra'abu El Naga, another team meticulously organized fragmented human remains, uncovering stories of women in ancient Egyptian society. American archaeologist Susanne Onstein explored the monumental building campaigns of Hetshepsood, whose colossal obelisks stood as testaments to her reign. Amidst pottery fragments and evidence of temples built by Hetshepsood and her stepson Tutmose III, the archaeologists faced personal reflections and challenges. The winds of history whispered through the sands, as each discovery brought them closer to unraveling the enigmatic legacy of Pharaoh Queen Hetshepsood. How do the discoveries and restoration efforts at Hatshepsut's temple and other archaeological sites in Egypt contribute to our understanding of her reign and the broader role of women in ancient Egyptian society? |
| Ground truth | Egypt, the richest source of archaeological treasures on the planet, beneath this desert landscape, why the secrets of this ancient civilization? Wow! You can see why a Pharaoh's chosen place, now, for a full season of excavations, our cameras have unprecedented access, follow teams on the front line of archaeology. I'm writing so fast because I'm subsided! It's an entrance, we can see an entrance. Revealing varied secrets. I have just been told that they have found something Making discoveries, they could rewrite ancient history. This time, new secrets about one of Egypt's greatest rulers, the Pharaoh Queen, Hatshepsut, doctors of Ranski discovers very treasures that her magnificent temple had reached was to be remembered for millions of years. The Darnels uncovered how she formed a mysterious double identity to seize power. For my beloved daughter, not son, and John and Maria, honor a rare and intriguing statue. |

Table 12: Comparison of LLM-generated teaser narrations—Example II

| Model | Narration |
|-------|-----------|
| Extractive | the mountains here were a battleground when US and NATO troops were still in the country. outside bombs and airstrikes have left their mark. the damage lines the road. in her large garden, Dr. Rochanak Wardak takes us down memory lane. gynecologist dr. rochanak's patients are still suffering the deprivation of that war. her shoulders aren't neck hurt. it makes no difference if I eat or not. with the Taliban, things are good now. it's peaceful. there was a lot of war and fighting. every pregnant woman wants to know the sex of her baby. yes, boy is power of the family. the drive through Kabul will reduce more drug users and street children than we've seen before. more than half of the population is hungry; food prices are soaring. women sit in front of bakeries, begging for a loaf of bread. but it's not enough to fill our stomachs. frida ghitis: the hunger crisis is on the government of the Taliban. she says they want a gender segregated society, but girls dare to dream of being able to do the same. aisha and her sister are trying to find their place in this Afghanistan. she says she has lost her spirit since the Taliban returned to power. aisha khan: if i were president, I would form an all-female government. she says women would be allowed to choose any job they wanted, study and live freely. her country is tired. tired of betrayal. you are songless and voiceless. cnn's aisha nasri meets with teen who wants to join the Taliban to serve the Islamic Emirate. she says many Afghans fear Taliban rule, but he and his friends brush aside such concerns, she adds. they say talk of explosions, targeted killings and arbitrary arrests is nothing more than Western propaganda. the traditional pottery of Istarlif was so foreign as darling when u.s. troops were still in the country. the pressure on visitors has affected the potters of the mountain village, which is now closed to tourists and a monopoly by the taliban. if no one buys their wares, they may not be able to do what they have always done. "we're afraid of new wars and more hardship," says Abdul manan. he says the traditional life has survived much, but now feels under threat again. |
| LLM-generated | We traveled west from Kabul to the war-torn province of Maidan-Wardak to meet Dr. Rochanak Wardak, a former politician and Taliban supporter, who reminisces about her efforts to build a new Afghanistan governed by parliament. War devastated the country, halting its development and education, leading Dr. Rochanak, a gynecologist who now runs a small home clinic, to witness her patients still suffering from the war's deprivation. With the cessation of fighting, travel has become safer, and while the region is now peaceful under the Taliban, there remains a cultural emphasis on the importance of having boys for family power and security, with some boys she helped deliver now becoming fighters. Dr. Rosha Nakh, who works for the Taliban overseeing female and maternal health, hopes for a change in their stance on women's activities, but faces the harsh reality of a male-dominated regime and widespread poverty in Kabul. The Taliban blame the hunger crisis on international sanctions and aid cuts, asserting their right to enforce their vision of Islamic rule and demanding autonomy, while their policies focus more on controlling women's lives than addressing economic issues. The Taliban's ban on secondary education for girls has left young women like Aisha, who once aspired to be doctors and pursued careers, feeling lost and hopeless as they struggle to find their place in the new Afghanistan. Aisha feels like a prisoner and blames the West for abandoning Afghanistan to the Taliban, which has led her to withdraw from public life and dream of an all-female government to ensure women's freedom and opportunities. The men would stay at home to experience the restrictive life of Afghan women, while Isatullah, a former farmer turned Taliban member, dismisses fears of Taliban rule as Western propaganda and expresses his support for the Sharia-based Islamic Emirate. The Taliban's restrictions on visitors, especially women, have severely impacted the traditional pottery business in the mountain village of Istarlif, leaving craftsmen like Abdul Manan and his son Ramish struggling to sustain their family craft. Abdul Manan, a traditional artist, expresses fear for his livelihood and safety amidst ongoing conflicts and deserted marketplaces, highlighting the persistent threat to their peaceful existence despite not taking sides in wars. |
| LLM-generated (tuned) | The journey west from Kabul to the war-torn province of Maidan-Wardak was fraught with memories of conflict and hope. Dr. Rochanak Wardak, a former politician and Taliban supporter, welcomed us into her small home clinic. Once a gynecologist with dreams of a parliamentary Afghanistan, Dr. Rochanak now witnesses the lingering scars of war in her patients' eyes. The cessation of fighting has brought a fragile peace, but the cultural emphasis on having boys for family power remains. Some of the boys Dr. Rochanak helped deliver are now fighters, a stark reminder of the cycle of conflict. Dr. Rosha Nakh, working for the Taliban in female and maternal health, dreams of a shift in the regime's stance on women's activities. However, the male-dominated reality and widespread poverty in Kabul cast a long shadow over her hopes. The Taliban blame international sanctions for the hunger crisis, focusing more on controlling women's lives than addressing economic issues. Young women like Aisha, once aspiring doctors, now feel lost and hopeless, their dreams crushed by the Taliban's ban on secondary education for girls. In the mountain village of Istarlif, traditional potter Abdul Manan and his son Ramish struggle to sustain their family craft amidst severe restrictions on visitors. Abdul Manan's fear for his livelihood and safety underscores the persistent threat to their peaceful existence, a poignant reminder of the ongoing conflicts that continue to shape Afghanistan's future. How does Dr. Rochanak Wardak reconcile her support for the Taliban with her aspirations for a progressive Afghanistan, especially in light of the Taliban's restrictive policies on women's education and activities? |
| Ground truth | Taliban fighters rule the streets, armed with weapons once used by their enemies. We see them all over Kabul. The Taliban's return to power has changed millions of lives, especially the lives of Afghan women. I feel like a prisoner, nothing is in my own hands anymore, before I could do whatever I wanted. Of course, I'm very much happy. You know, the head is no war. We are living like other people in the world. The Taliban are back and so is their ideology, but what's it like living in today's Afghanistan? |

## L  RESULTS FOR THE BASELINE VIDEO SUMMARIZATION MODELS

### L.1  BASELINE MODELS

We include the following two video summarization models as our baseline models and report results in Table 16. We observe that TeaserGen achieves a relatively higher F1 score and also provides narrations with higher audio-visual correspondence that are effective in engaging the audience.

- **CSTA** (Son et al., 2024): This baseline model proposes a CNN-based SpatioTemporal Attention (CSTA) to summarize videos. Note that CSTA cannot process videos with more than 5,000 frames, so we chunked the long videos and recombined them after processing.
- **A2Summ**(He et al., 2023a):This baseline model proposes a unified multimodal transformer-based model to extract the most important information from different modalities. A2Summ cannot handle the long documentaries in our dataset, so we divide them into 10 chunks and recombine them afterward.

Table 16: Objective evaluation results with LLM-generated narrations.

| Model | Query | Decoding | DP | F1 (%)↑ | REP (%) | SCR (%) | CLIPScore | VTGHLS |
|---|---|---|---|---|---|---|---|---|
| **Baseline models** | | | | | | | | |
| Random | Random | - | - | 1.67 | 4.05 | 7.81 | 0.56 | 0.75 |
| CLIP-NN | Narration | Greedy | × | 0.11 | 92.73 | 8.29 | 0.69 | 0.79 |
| UniVTG (2023b) | Title | Rank | - | 1.82 | 0 | 89.68 | 0.58 | 1.01 |
| CLIP-it (2021b) | Narration | Rank | × | 1.24 | 0 | 99.39 | 0.56 | 0.61 |
| CSTA(2024) | - | - | × | 1.78 | 0 | 5.00 | - | - |
| A2Summ(2023a) | - | - | × | 0.77 | 0 | 95.23 | 0.54 | 0.61 |
| **Pretraining-based models** | | | | | | | | |
| TeaserGen-PT | Title | Thresholding | - | 1.85 | 0 | 13.16 | 0.56 | 1.02 |
| TeaserGen-PT | Narration | Thresholding | - | 1.07 | 21.38 | 22.58 | 0.58 | 1.45 |
| TeaserGen-PT-CLIP | Narration | Threshold | × | 1.31 | 27.23 | 24.10 | 0.58 | 0.74 |
| **Learning-based models** | | | | | | | | |
| TeaserGen-LR | Narration | Greedy | × | 1.56 | 31.97 | 27.18 | 0.58 | 0.74 |
| TeaserGen-LR | Narration | Greedy | ✓ | 1.38 | 26.83 | 35.48 | 0.62 | 0.78 |
| TeaserGen-LR | Narration | Beam search | × | **1.88** | 24.16 | 41.97 | 0.58 | 0.74 |
| TeaserGen-LR | Narration | Beam Search | ✓ | **1.88** | 19.39 | 46.56 | 0.63 | 0.77 |
| Ground truth | - | - | - | 100 | >7.86 | 27.6 | 0.58 | 0.64 |

## M  ZERO-SHOT EVALUATION

We present the objective evaluation results in Table 17. We observe that TeaserGen achieves a relatively higher F1 score and also provides narrations with audio-visual correspondence that are effective in engaging the audience. Zero-shot samples can be found on our demo page [1].

Table 17: Objective evaluation on Zero-Shot Examples

| Model | REP (%) | SCR (%) | CLIPScore | VTGHLS |
|---|---|---|---|---|
| CSTA(2024) | 4.90 | 5.40 | - | - |
| A2Summ(2023a) | 0 | 96.25 | 0.54 | 0.57 |
| TeaserGen-PT | 0 | 13.95 | 0.58 | 0.99 |
| TeaserGen-LR | 20.3 | 48.69 | 0.62 | 0.87 |

## N  ABLATION STUDY ON MINIMUM CLIP LENGTH AND OVERLAPPING TOLERANCE

During our experiments, we aimed to avoid having clip lengths that were too short, as this would make the output fragmented. However, setting the minimum clip length too high would exclude too many clips, thereby sacrificing audio-visual correspondence. We conducted an ablation study on different combinations, as shown in the Table 18 for TeaserGen-PT with narration being the query. We observe that selecting 3 as our minimum clip length results in a scene change rate that is mostly close to the ground truth and provides higher audio-visual correspondence.

Table 18: Experiments on Minimum Clip Length and Overlapping Tolerance

| Min Clip Length | Overlapping Tolerance | F1(%) | REP (%) | SCR (%) | CLIPScore | VTGHLS |
|:---:|:---:|:---:|:---:|:---:|:---:|:---:|
| 1 | 0 | 0.97 | 20.19 | 48.12 | 0.57 | 1.45 |
| 2 | 0 | 0.98 | 19.80 | 29.39 | 0.57 | 1.45 |
| 3 | 0 | 0.96 | 19.96 | 22.49 | 0.57 | 1.45 |
| 3 | 1 | 1.07 | 21.38 | 22.58 | 0.58 | 1.45 |
| 5 | 1 | 1.14 | 21.01 | 15.33 | 0.57 | 1.44 |
| 5 | 3 | 1.08 | 26.32 | 15.44 | 0.57 | 1.44 |
| 10 | 5 | 1.16 | 26.82 | 8.38 | 0.57 | 1.41 |

## O    DETAILS OF SEARCH ALGORITHM

The goal of the proposed searching algorithm is to maximize the total video-narration correspondence while maintaining the desired length. The desired length is defined as the duration of the synthesized speech for each sentence. To facilitate smooth transitions between concatenated video clips, we add a one-second buffer time between sentences to ensure that the length of the audio track length is greater than or equal to that of the visual content. For each sentence in the documentary, we generate a score curve (with the x-axis representing video time and the y-axis representing the score). This curve reflects frame importance (defined by its likelihood of being included as a highlight in video highlight detection) and text-visual correspondence to the sentence. During the search process, the algorithm selects a threshold such that the intersections of the score curve and the threshold determine the start and end times of video clips. These clips are chosen based on the condition that their scores exceed the threshold. Multiple clips can satisfy this condition. The search begins at the top of the score curve, lowering the threshold incrementally (by 0.001 steps) until the total length of the selected clips matches the duration of the generated audio waveform. This final threshold is considered optimal, as illustrated in Figure 2. For each sentence in a given documentary, we associate it with a score curve. The sentence-level VTGHLS is calculated by averaging the VTGHLS values of all frames fall in that sentence whose VTGHLS exceeds the threshold in the corresponding score curve. Finally, we compute the overall average VTGHLS across all sentences in all documentaries within the test set. To improve computational efficiency, we take advantage of the monotonic increase in the total length of selected video clips as the threshold decreases. Instead of linearly decreasing the threshold, a binary search algorithm is applied to quickly identify the optimal threshold.

The estimated VTGHLS value of 0.64 for ground truth is used in Table 3 is only for comparison purpose and it is not used anywhere in our training or inference pipeline.

