# OpenReview forum: "TeaserGen: Generating Teasers for Long Documentaries"
_ICLR.cc/2025/Conference — ICLR 2025 Poster_

### Official Review · Reviewer_Hqts · 2024-10-30

**Soundness:** 3
**Presentation:** 2
**Contribution:** 2
**Rating:** 5
**Confidence:** 4

**Summary:**

This paper presents TeaserGen, a two-stage system for generating promotional teasers for long documentaries. Leveraging a their proposed dataset, DocumentaryNet, the authors aim to generate teasers by first synthesizing teaser narrations from documentary transcripts using a large language model. They then use a language-vision model to select relevant visual content that aligns with the narration. In that process, to avoid repeat frames, they proposed some methods to alleviate. The study compares a pretraining-based model (TeaserGen-PT) and a deep sequential learning model (TeaserGen-LR) for narration-video alignment. Experimental results show some advantages of the pretraining-based approach over directly trained autoregressive models.

**Strengths:**

1)The DocumentaryNet dataset of 1,269 documentary-teaser pairs fills a gap in the community by providing a publicly available resource for multimodal summarization and teaser generation, and the system shows potential for applications in media, advertising, and education for automated content promotion.

2)The inclusion of various evaluation metrics, including scene change rate and repetitiveness, provides a more nuanced assessment of the teaser’s quality.

**Weaknesses:**

1)The approach predominantly relies on pretrained language-vision models, limiting novelty. This reliance raises questions about the model’s true capacity to generate creative outputs, as it functions more as an information retrieval system instead of a real generative system.

2)The proposed model does not consider the alignment of audio cues like music or sound effects with visual elements, which limits its ability to produce emotionally engaging teasers, I've checked in your demo, there also exists this issue, and none of above baseline methods consider this aspect, and if you could consider this, that would be a great contribution in this domain. In this aspect, you get all frames by your frame-matching system driven by the text narration you got from llm, so the audio should also be different correspond to each frame in this proposed hypothesis. And by the way, this multi-stage method, means multi-stage information loss and also your pretrained model from general domain also have information loss, so I don't think this method work well for the task that you work with in this paper.

3)The sentence-by-sentence matching approach does not effectively capture scene continuity, leading to potentially fragmented visual sequences that lack coherence even though you utilized the smoothing and regularisation.

4)The experiments are mainly based on one proprietary dataset (other methods also didn't train on your dataset) and lack extensive ablation studies for elements such as the diffusion prior model and threshold sensitivity, which are critical to understanding the model’s flexibility and robustness.

**Questions:**

1)Line229-230: You should detail this part about how to find the threshold and does it sensitive and what is the pretrained model here. especially in the supplementary material.

2)Line267-268: Please illustate well about the diffusion prior and how it helps and also do you conduct the ablation study about this.

3)Line693-706: You collect these media from three main sources, but in the appendix A that you explain about the dataset, I didn't see any discussion about the copyright for these videos.

**Details Of Ethics Concerns:**

In the dataset part where they specify in the Appendix A, they didn't discuss about all three parts sourcees of their collected medias, the copyright and availability for research should be discussed.

---

> ### Author Response · Authors · 2024-11-21
>
> We appreciate the reviewer for the thoughtful review and constructive comments. Please find our response below.
>
> **[Weakness 1]**:
>
> We would like to highlight that we also introduce TeaserGen-LR, a learning-based end-to-end architecture designed for this task. In this paper, we focus on the descriptive and precise aspects of teaser generation and present several metrics to evaluate its performance. To ensure factual accuracy, particularly in educational documentaries, we advocate for extracting visual content rather than generating it. Creativity in our approach arises from engaging narration rather than visual content. Although this paper does not explore music or sound effects, our publicly available dataset, which spans multiple modalities, provides a solid foundation for future research in areas such as long-form video-to-audio generation[1,2,3] audio separation [4], and trailer generation [5]. In this paper, we limit our scope more on the non-creative aspects of the task of teaser generation in generating only the narration and the accompanying visuals, and we believe our proposed dataset can enable future research on modeling the creative aspects of this topic.
>
> **[Weakness 2]**:
>
> We thank the reviewer for raising this concern. As mentioned in the discussion section, we are interested in generating visually-aligned sound effects and background music as a next step. We also note that this is an active area of research [2, 6, 7]. Given the complexity of this task, we chose to focus on narration and visual generation in this paper as an initial step toward the ultimate goal of generating teasers with visually-aligned sound effects and background music. We evaluate the output from the first stage, i.e the generate narrations in our subjective test. Our subjective (see the attached table) test shows that generated narrations are coherent, informative, and engaging. In our approach, we sought to mitigate these losses by fine tuning the pretrained model in Table 9 (see the attached table). However, we recognize that these strategies may not fully eliminate the issue, therefore, we proposed a learning based method to directionally mapping narrations to visual contents. The F1 score of the learning-based approach TeaserGen-LR is higher than the F1 score of the interval-based approach TeaserGen-PT.
>
> | Narration       | Organization    | Informativeness | Engagingness |
> |-------------------------|-----------------|-----------------|-----------------|
> | Naive Summarization| 3.58 $\pm$ 0.57     | 3.72 $\pm$ 0.47     | 3.60 $\pm$0.56    |
> | Finely-tuned Scripts| 3.88 $\pm$ 0.44     | 3.82  $\pm$0.54     | 3.70 $\pm$ 0.46 |
>
> Table 9:
>
> | Query         | Decoding      | Finetune | F1 (%) | REP (%) | SCR (%) | CLIPScore | VTGHLS |
> |---------------|---------------|----------|--------|---------|---------|-----------|--------|
> | Title         | Thresholding  | No       | 1.85   | 0       | 13.16   | 0.56      | 1.02   |
> | Narration     | Thresholding  | No       | 1.07   | 21.38   | 22.58   | 0.58      | 1.45   |
> | Narration     | Thresholding  | Yes      | 1.23   | 41.39   | 24.01   | 0.56      | 1.05   |
>
> **[Weakness 3]**:
>
> We observe that scene changes occur more frequently "within sentences" in the ground truth teaser, and thus we leverage the narration to ensure continuity and coherence of the narrative in the proposed two-stage system. On average, there are 2.84 scene changes per sentence, meaning scene changes typically happen within sentences. To ensure the flow of generated scripts, we design careful prompts. For example, we convert the summary of each chunk into a story using prompts such as: “Rewrite the paragraph into an engaging story opening in 10 sentences or less, keeping all names and avoiding the use of pronouns.” Our subjective test in Section 5.5 (see the attached table) demonstrates that our generated narrations are coherent, informative, and engaging.
>
> | Narration       | Organization    | Informativeness | Engagingness |
> |-------------------------|-----------------|-----------------|-----------------|
> | Naive Summarization| 3.58 $\pm$ 0.57     | 3.72 $\pm$ 0.47     | 3.60 $\pm$0.56    |
> | Finely-tuned Scripts| 3.88 $\pm$ 0.44     | 3.82  $\pm$0.54     | 3.70 $\pm$ 0.46 |
>
> For the narration-visual matching process, to avoid frequent scene changes and overly-fragmented visuals, we have proposed an interval-based binary search algorithm in TeaserGen-PT that does not allow selecting video clips shorter than 3 sec.

---

> ### Author Response · Authors · 2024-11-21
> **Continued**
>
> **[Weakness 4 and Question 2]**:
>
> We would like to highlight that we conducted an ablation study on the diffusion prior in Section 5.6, Table 6 (attached below). Our ablation study indicates that applying the diffusion prior leads to lower repetitiveness and higher language-vision correspondence, as reflected by higher CLIP and VTGHLS scores. The decreased repetitiveness may result from the sampling process in the diffusion prior, which is a generative model that produces diverse image embeddings for the same input text embeddings. The increased CLIPScore likely stems from the diffusion prior's ability to bridge the gap between textual embeddings and visual embeddings. Comparing the two results of TeaserGen-LR using beam search decoding, we also observe that applying the diffusion prior results in a higher scene change rate, possibly because the diffusion prior further encourages scene transitions.
>
> | Model          | Decoding     | DP   | Smoothing | F1 (%) | REP (%) | SCR (%) | CLIPScore | VTGHLS |
> |----------------|--------------|------|-----------|--------|---------|---------|-----------|--------|
> | TeaserGen-LR   | Greedy       | No   | No        | 0.53   | 81.32   | 27.38   | 0.59      | 0.74   |
> |    TeaserGen-LR            | Greedy       | No   | Yes       | 1.56   | 1.56    | 31.97   | 0.58      | 0.74   |
> |    TeaserGen-LR            | Greedy       | Yes  | Yes       | 1.38   | 26.83   | 35.47   | 0.62      | 0.78   |
> |     TeaserGen-LR           | Beam Search  | No   | Yes       | 1.88   | 24.16   | 41.97   | 0.58      | 0.74   |
> |    TeaserGen-LR            | Beam Search  | Yes  | Yes       | 1.88   | 19.39   | 46.56   | 0.63      | 0.77   |
>
> To demonstrate the robustness of our model, we also report zero-shot examples on TED Talks and old movies, which are showcased on our demo page (https://54321anonymous.github.io/ICLR2025/). Our training samples include documentaries from various genres, such as nature scenes and social events. We observe that the model can **effectively generate teasers for TED Talks and old movies, even though it has never been trained on these domains**, highlighting the robustness of the proposed methods. Below are the objective results of the zero-shot examples. We further compare TeaserGen-PT and TeaserGen-LR with two extra video summarization models: CSTA[9] and A2Summ[10] in zero-shot scenarios in the table below.
>
> | Model           | REP (%) | SCP (%)  | CLIPScore | VTGHLS |
> |---------------|---------|----------|-----------|--------|
> | CSTA          | 4.9     | 5.4     | N/A       | N/A    |
> | A2Summ | 0 | 96.25 | 0.54 | 0.57 |
> | TeaserGen-PT     | 0       | 13.95  | 0.58      | 0.99   |
> | TeaserGen-LR    | 20.3    | 48.69    | 0.62      | 0.87   |
>
> **[Weakness 4 & Question 1]**:
>
> We utilize a video temporal grounding pretraining model, UniVTG [8], to generate a score curve that measures frame importance (defined by its likelihood of being included as a highlight in video highlight detection) in addition to text-visual correspondence. For each sentence $s_i$​, we generate audio using a pretrained text-to-speech model (https://platform.openai.com/docs/guides/text-to-speech).
> Our objective is to determine a threshold on the score curve, where the x-axis represents video time, and the y-axis represents the score. The goal is to select the start and end times of video clips such that the scores within these clips exceed the threshold. Multiple clips may satisfy this criterion, and we ensure that the total length of the selected clips matches the duration of the generated audio. Furthermore, we ensure each selected clip exceeds a minimum length and resolve overlaps with preceding sentences. As the threshold is lowered, the total length of the selected video clips increases monotonically. To maintain computational efficiency, we employ a binary search to determine the optimal threshold.
>
> **[Question 3]**:
>
> Thank you for pointing that out! We recognize the importance of addressing potential risks associated with copyrighted content, as noted in our ethics statement:
> "Our generative model, trained on copyrighted material, carries a risk of generating content that could potentially infringe on copyright."
> To address this concern, we have taken several steps to mitigate potential risks. First, we do not release raw video data from the dataset to avoid unauthorized redistribution of copyrighted content. Instead, we provide publicly accessible YouTube links along with our annotations, ensuring that users can access the data within the bounds of the original platform's terms of service.
> Furthermore, our use of the dataset complies with fair use principles, focusing on research purposes such as analysis, annotation, and model development. We are also open to feedback on additional steps we can take to improve transparency and ethical considerations in this area.

---

> ### Author Response · Authors · 2024-11-21
> **Reference**
>
> [1]: Polyak et al., Movie Gen: A Cast of Media Foundation Models, ECCV, 2020 (https://arxiv.org/abs/2410.13720).
>
> [2]: Tian et al., VidMuse: A Simple Video-to-Music Generation Framework with Long-Short-Term Modeling, Arxiv, 2024(https://arxiv.org/abs/2406.04321).
>
> [3]: Li et al., MuVi: Video-to-Music Generation with Semantic Alignment and Rhythmic Synchronization, Arxiv, 2024 (https://arxiv.org/abs/2410.12957).
>
> [4]: Huang et al., High-Quality Visually-Guided Sound Separation from Diverse Categories, Arxiv, 2024 (https://arxiv.org/abs/2308.00122).
>
> [5]: Argaw et al., Towards automated movie trailer generation, CVPR, 2024 (https://arxiv.org/abs/2404.03477.)
>
> [6]: Kang et al., Video2Music: Suitable music generation from videos using an Affective Multimodal Transformer model, Expert Systems with Applications, Arxiv, 2023 (https://doi.org/10.1016/j.eswa.2024.123640).
>
> [7]: Lee et al., Video-Foley: Two-Stage Video-To-Sound Generation via Temporal Event Condition for Foley Sound, Arxiv, 2024 (https://arxiv.org/abs/2408.11915).
>
> [8]: Lin et al., UniVTG: Unified Video-Language Temporal Grounding, ICCV, 2023 (https://arxiv.org/abs/2307.16715)
>
> [9]: Son et al., CSTA: Spatiotemporal Attention for Video Summarization, CVPR, 2024 (https://arxiv.org/abs/2405.11905)
>
> [10]: He et al., Align and Attend: Multimodal Summarization with Dual Contrastive Losses, CVPR, 2023 (https://arxiv.org/abs/2303.14150)

---

### Official Review · Reviewer_VbpS · 2024-10-31

**Soundness:** 3
**Presentation:** 3
**Contribution:** 3
**Rating:** 3
**Confidence:** 5

**Summary:**

- The paper presents TeaserGen, a two-stage system for creating teasers from long documentaries, addressing challenges like audiovisual alignment, smooth transitions, and factual accuracy. To support this, the authors developed DocumentaryNet, a dataset of 1,269 documentaries with teasers, including multimodal elements like video, narration, and sound effects.

- TeaserGen first generates teaser narration from the documentary’s transcript using a large language model, creating an engaging summary. It then pairs visuals with narration through either a pre-trained contrastive language-vision model or a deep sequential model to match visuals accurately.

- Results show that TeaserGen outperforms baseline models in maintaining coherence and alignment, offering a streamlined approach to automated teaser generation. DocumentaryNet and TeaserGen together provide valuable tools for advancing multimodal content modeling in documentary summarization.

**Strengths:**

- Tackles a unique problem in automated teaser generation for documentaries with TeaserGen, a creative, narration-centered two-stage approach that combines large language models with language-vision models for cohesive narration and visual alignment, showing effective and innovative use of existing technologies.

- Provides solid empirical support with comparisons to baseline models across objective (e.g., F1 score, CLIPScore) and subjective metrics, as well as the introduction of DocumentaryNet, a multimodal dataset with documentary-teaser pairs that enriches resources available for this research area.

- The work has significant potential impact by addressing a real-world gap in video summarization for documentary-style content, with applications in multimedia and educational fields, and establishes a foundation for further multimodal research, likely to stimulate new directions in long-form video modeling.

**Weaknesses:**

- The paper presents an innovative approach to video teaser generation using pretrained language-vision models, but several issues need to be addressed to enhance its clarity and robustness. In rows 210 and 211, there is a notation inconsistency where $S$ is defined as a sequence of language tokens, yet later each $S_i$ is referred to as a waveform (audio signal). This inconsistency creates confusion, and it's crucial for the notation to consistently represent either language tokens or audio waveforms throughout the paper to avoid misunderstandings.

- In Section 4.2.1, the method relies heavily on a single pretrained VTGHS model without sufficient ablation studies or comparisons with alternative architectures, which weakens the validation of the approach. The constraints imposed such as a minimum clip length of three seconds and a one-second overlap between clips, appear arbitrary and lack theoretical or empirical justification, raising questions about their effectiveness and impact on the results. Additionally, using a frame rate of only one frame per second (1 FPS), as mentioned in rows 453 to 455, is inadequate for videos that change dynamically. This low frame rate hinders the model's ability to effectively capture motion and make accurate predictions.

- In Section 4.2.2, the model extracts features at a low frame rate of 1 FPS, causing multiple frames to share identical sentence embeddings. This coarse temporal resolution fails to capture dynamic changes within the video, leading to overly repetitive scenes. The absence of fine-grained temporal annotations in the dataset prevents the model from effectively distinguishing and assigning unique embeddings to semantically similar frames occurring at different times. As a result, the approach struggles to maintain diversity and temporal coherence in the generated visual content.

- Regarding threshold selection, in row 320, the paper states, `We estimate a VTGHLS of 0.64 for the ground truth teasers`, but there is insufficient explanation about the criteria used to select this threshold value. The paper does not analyze how varying this threshold affects the results. Since there are ablation studies on changing the matching score function, as mentioned in Section 5.6, it is necessary to explore this threshold value in more detail to understand its impact on the overall pipeline.

- Table 4 may present an unfair comparison because the TeaserGen models utilize advanced decoding techniques like beam search, which can enhance performance, while the baseline models do not use these techniques. For a fair comparison that accurately reflects each model's true capabilities, all models should employ similar decoding methods.

- A significant limitation in the methodology is the heavy reliance on subjective metrics without incorporating standardized quantitative measures, as mentioned in rows 468 to 470. This over-reliance weakens the study's reproducibility and generalizability. While subjective listening tests provide valuable insights, the absence of automatic evaluation metrics commonly used in natural language processing and computer vision (such as ROUGE, BLEU, BERTScore, or perplexity for the generated narration tex) makes it difficult to objectively compare the results with other approaches or validate the findings across different contexts.

- The TeaserGen-LR model uses a limited architecture with only three transformer layers, which may not capture complex patterns as effectively as the diffusion prior's more extensive 12-block backbone. Relying solely on L2 distance as the loss function might not fully capture perceptual similarities, leading to less nuanced image generation. Additionally, training the model for only 15 epochs on a small test set of 49 documentaries raises concerns about underfitting and limits the generalizability of the results.

- The study does not discuss the potential computational overhead introduced by incorporating the diffusion prior, which could affect the practicality and scalability of the approach. While higher scene change rates may increase visual diversity, they might also compromise the narrative coherence of the generated teasers. Addressing these issues would enhance the study's robustness and applicability.

**Questions:**

- Could you clarify the notation used in rows 210 and 211? Specifically, is  $S$  intended to represent a sequence of language tokens or audio waveforms? Consistent notation throughout the paper would enhance understanding and prevent confusion.

- Have you considered evaluating alternative architectures or conducting ablation studies to assess the robustness of relying solely on the pretrained VTGHL model? Exploring different models could strengthen the validation of your approach.

- What is the rationale behind setting the minimum clip length to three seconds and the overlap between clips to one second? Providing theoretical or empirical justification for these specific constraints would help in understanding their impact on the results.

- Given that a frame rate of 1 FPS may be insufficient for capturing dynamic video content, have you experimented with higher frame rates? How does increasing the frame rate affect the model's ability to capture motion and improve prediction accuracy?

- How does your model address the issue of repetitive scenes arising from multiple frames sharing identical sentence embeddings? Have you explored methods to incorporate fine-grained temporal information or annotations to enhance diversity and temporal coherence?

- Could you elaborate on how the VTGHLS threshold of 0.64 was determined? Additionally, have you investigated how varying this threshold influences the results, perhaps through an ablation study?

-  Given that the TeaserGen-LR model uses only three transformer layers, have you tested deeper architectures to see if they capture complex patterns more effectively? Would increasing the number of layers improve performance?

- Have you considered using standardized quantitative evaluation metrics like ROUGE, BLEU, BERTScore, or perplexity for assessing the generated narration text? Including these metrics could enhance the reproducibility and comparability of your study.

- Relying solely on L2 distance as the loss function might not fully capture perceptual similarities. Have you experimented with alternative loss functions, such as perceptual loss, SSIM loss and so on to potentially achieve more nuanced image generation?

- Considering that the models were trained for only 15 epochs on a relatively small test set of 49 documentaries, have you explored training for more epochs or using a larger dataset? How might this affect model performance and generalizability?  To test the generalizability of your approach, have you considered evaluating the model on additional datasets beyond the 49 documentaries? How does the model perform on different genres or types of video content?

- Could you provide details on the computational resources required by the diffusion prior model? Understanding the computational overhead would help assess the practicality and scalability of your approach in real-world applications.

- Higher scene change rates may enhance visual diversity, have you evaluated their impact on the narrative coherence of the teasers? How do you balance diversity with maintaining a coherent and engaging storyline?

- You employ different CLIP models (CLIP-ViT-B/32 and CLIP-ViT-L/14) for different components of your system. Have you considered the potential inconsistencies this might introduce? Would using the same CLIP model throughout improve the alignment between textual and visual data?

-How does the frame extraction rate influence the model's performance in terms of capturing essential visual information? Have you analyzed the trade-offs between computational efficiency and the richness of visual features at different frame rates?

---

> ### Author Response · Authors · 2024-11-20
>
> We appreciate the reviewer for the thoughtful review and constructive comments. Please find our response below.
>
> **[Weakness 1 & Question 1]**:
>
> Thank you for pointing that out! $s_{i}​$ is a sequence of language tokens, and S = {$s_{1}$, $s_{2}$ ... $s_{m}$} represents a sequence of sequences of language tokens. We use TTS to generate the audio waveform for each sentence and denote $\tau_{i}$ as the length of the generated audio waveform for the sequence of language tokens $s_{i}$
>
> **[Weakness 2 & Question 2]**:
>
> Yes. We conducted an **ablation study on matching score function** in section 5.6 and also proposed **a learning based approach** TeaserGen-LR that leverages transformer architecture. We conduct ablation study on TeaserGen-LR by experimenting with decoding strategy and diffusion prior module.
>
> In Table 3(we attached table below), we report the objective result of using CLIPScore as the matching metric.
>
> | Model            | Query     | Decoding    | Matching Score Function | F1 (%)  | REP (%)   | SCR (%)   | CLIPScore | VTGHLS |
> |-------------------|-----------|-------------|------------------------|------|-------|--------|-----------|--------|
> | TeaserGen-PT      | Title     | Thresholding| VTGHLS                 | 1.85 | 0     | 13.16  | 0.56      | 1.02   |
> | TeaserGen-PT      | Narration | Thresholding| VTGHLS                 | 1.07 | 21.38 | 22.58  | 0.58      | 1.45   |
> | TeaserGen-PT-CLIP | Narration | Thresholding| CLIPScore              | 1.31 | 27.23 | 24.10  | 0.58      | 0.74   |
> | Ground Truth      |           |             |                        | 100  | >7.86 | 27.6   | 0.58      | 0.64   |
>
> In Table 3 (see the table below), we also report the ablation study on TeaserGen-LR, experimenting with the decoding strategy and the diffusion prior module.
>
> | Model        | Query     | Decoding    | Diffusion Prior | F1 (%)  | REP (%)  | SCR  (%)  | CLIPScore | VTGHLS |
> |--------------|-----------|-------------|-----------------|------|-------|--------|-----------|--------|
> | TeaserGen-LR | Narration | Greedy      | No              | 1.56 | 31.97 | 27.18  | 0.58      | 0.74   |
> | Teaser-LR    | Narration | Greedy      | Yes             | 1.38 | 26.83 | 35.48  | 0.62      | 0.78   |
> | Teaser-LR    | Narration | Beam Search | No              | 1.88 | 24.16 | 41.97  | 0.58      | 0.74   |
> | Teaser-LR    | Narration | Beam Search | Yes             | 1.88 | 19.39 | 46.56  | 0.63      | 0.77   |
> | Ground Truth |           |             |                 | 100  | >7.86 | 27.6   | 0.58      | 0.64   |
>
> **[Weakness 2 & Question 3]**:
>
> During our experiments, we aimed to avoid having clip lengths that were too short, as this would make the output overly fragmented. However, setting the minimum clip length too high would exclude too many clips, thereby sacrificing audio-visual correspondence. **We conducted an ablation study on different combinations in the following table for TeaserGen-PT with textual narration being the query**. We observe that selecting 3 as our minimum clip length and setting overlapping tolerance being 1 results in a scene change rate that is closest to the ground truth and provides higher audio-visual correspondence.
>
> | Minimum Clip Length | Overlapping Tolerance | F1 (%)  | REP (%)  | SCR (%)   | CLIPScore | VTGHLS |
> |---------------------|-----------------------|------|-------|--------|-----------|--------|
> | 1                   | 0                     | 0.97 | 20.19 | 48.12  | 0.57      | 1.45   |
> | 2                   | 0                     | 0.98 | 19.8  | 29.39  | 0.57      | 1.45   |
> | 3                   | 0                     | 0.96 | 19.96 | 22.49  | 0.57      | 1.45   |
> | 3                   | 1                     | 1.07 | 21.38 | 22.58  | 0.58      | 1.45   |
> | 5                   | 1                     | 1.14 | 21.01 | 15.33  | 0.57      | 1.44   |
> | 5                   | 3                     | 1.08 | 26.32 | 15.44  | 0.57      | 1.44   |
> | 10                  | 5                     | 1.16 | 26.82 | 8.38   | 0.57      | 1.41   |
> | Ground Truth        |                       | 100  | >7.86 | 27.6   | 0.58      | 0.64   |
>
> **[Weakness 3 & Question 4]**:
>
> No, we did not experiment with higher frame rates due to limited computing resources. We follow LfVS[1] and UniVTG[2] to use fps=1, and we find it to be working well in practice. Moreover, from Figure 1, we can clearly observe that a frame rate of 1 fps is sufficient to capture most scene changes. We believe that by increasing the frame rate, we will be able to capture better motion and achieve prediction accuracy, but it would also introduce some optimization challenges as well.

---

> ### Author Response · Authors · 2024-11-20
> **Continued**
>
> **[Weakness 3 & Question 5]**:
>
> Yes, we propose different methods to address the issue of repetitive scenes for TeaserGen-PT and TeaserGen-LR and enhance diversity and temporal coherence.
> 1) TeaserGen-LR: In section 4.2.2, we **incorporate diffusion prior to encourage diversity** within one sentence. In addition, diffusion prior helps to bridge the gap between textual embedding and frame embeddings. The transformer model can capture temporal coherence.
> 2) In addition, for the proposedTeaserGen-PT model, Our interval based approach aims to **select intervals(video clips) by thresholding** for each sentence. Therefore, no repeated frames will be selected within one sentence embedding.
>
> **[Weakness 4 & Question 6]**:
>
> The estimated VTGHLS value of 0.64 on the ground truth is computed as follows: We first match the ground truth teaser back to body contents. To determine the lowest distance that can be considered as close, we randomly pick 15 documentaries with teaser and body content frame pairs. We calculate the 20% L2 distances between each teaser frame and each body frame and consider it as the lowest distance threshold. We then extract the CLIP feature and select top 20 images with the highest cosine similarity to find those close in semantic meanings. Then we calculate the pixel-by-pixel L2 distance between the teaser frame and those 20 images. We pick the one with the lowest distance and smaller than the selected threshold.  Then we leverage a pretrained VTGHL model to calculate the audio-visual correspondence between matched frames(in body contents) and corresponding textual narrations plus the importance of the frame.
> Moreover, we would like to point out that **the optimal threshold for TeaserGen-PT is determined using binary search to maximize visual-textual narration correspondence while maintaining the desired video length**. When the threshold is lowered, the total length of the selected video clips increases monotonically. We leverage binary search to ensure computational efficiency.
>
> **[Weakness 5]**:
>
> We would like to point out that **decoding techniques like beam search are proposed module** and we conduct ablation study on decoding method in table 3(We attached table below). An advanced decoding method, such as beam search, results in a higher F1 score and reduced repetitiveness.
>
> | Model        | Decoding    | F1 (%) | REP (%) | SCR (%) | CLIPScore | VTGHLS |
> |--------------|-------------|--------|---------|---------|-----------|--------|
> | TeaserGen-LR | Greedy      | 1.56   | 31.97   | 27.18   | 0.58      | 0.74   |
> | TeaserGen-LR | Beam Search | 1.88   | 24.16   | 41.97   | 0.58      | 0.74   |
> | Ground Truth |             | 100    | >7.86   | 27.6    | 0.58      | 0.64   |
>
> **[Weakness 7 & Question 7]**:
>
> We believe that using a deeper and more advanced transformer architecture would likely lead to a better performance in capturing complex visual and temporal patterns. However, due to the limited computing budget, we did not experiment with different network architecture in this work.
>
> **[Weakness 6 & Question 8]**:
>
> We conduct **a subjective evaluation of our generated narration** in Section 5.5 by assessing the organization, informativeness and engagingness on a Likert scale from 1 to 5. Our carefully designed prompts are considered consistent, informative, and engaging by human evaluators. As mentioned in section 2, teaser generation is unlike traditional summarization tasks, teaser generation requires handling longer input sequences, higher compression rates, and ensuring the textual narration remains engaging and story-driven.
>
> | Narration | Organization | Informativeness | Engagingness |
> |----------|----------|----------|----------|
> | Naive Summarization  | 3.58 $\pm$0.57  | 3.72 $\pm$0.47 | 3.60 $\pm$ 0.56|
> |Finely-tuned scripts| **3.88 $\pm$ 0.44** | **3.82 $\pm$0.54**| **3.70 $\pm$0.46**|
>
> We report **common text summarization evaluation metrics**, specifically the average F1 score of ROUGE, in the table below. We compare our LLM-generated narration with the audio transcription of the ground truth teaser, which includes dialogues. Additionally, for reference, we compare the performance of TeaserGen narration with fully extractive narration from the main content and a naïve summarization approach.
>
> | Evaluation | Naive Summarization | Fully Extraction | TeaserGen |
> |----------|----------|----------|----------|
> | ROUGE-1 | 0.14  | 0.15 | 0.14|
> | ROUGE-2 | 0.02 | 0.02 | 0.02|
> | ROUGE-l | 0.12 | 0.14 | 0.12|

---

> ### Author Response · Authors · 2024-11-20
> **Continued**
>
> **[Question 9]**:
>
> We follow  TGT[4] and LfVS [1] to use L2 loss in the embedding space.  We are mapping textual embedding to image embedding with transformer architecture. The L2 loss is the MSELoss between predicted image embedding and ground truth embedding rather than in the pixel space.
>
> We would like to point out that we did not frame the textual narration-visual matching task as a generative task. Rather, we framed it as a retrieval task where we aim to retrieve the most relevant frames to accompany the LLM-generated textual narrations.
>
> **[Weakness 7 & Question 10]**:
>
> Here are the details of dataset splitting:
>
> | Split      | Number of Samples | Total Hours |
> |------------|--------------------|-------------|
> | Train      | 1026              | 514         |
> | Validation | 57                | 32          |
> | Test       | 49                | 29          |
>
> Our training samples include documentaries from various genres, such as nature scenes and social events. We also report zero-shot examples on TED Talks and old movies, which are showcased on our **demo page**(https://54321anonymous.github.io/ICLR2025/). We report the objective evaluation results on zero-shot examples in the table below. We observe that the models can **effectively generate teasers for TED talks and old movies as well**, while the models have never been trained on these domains, which showcases the robustness of the proposed methods.
>
> | Model        | REP (%) | SCP (%)  | CLIPScore | VTGHLS |
> |--------------|------|---------|----------|-----------|
> | TeaserGen-PT  | 0       | 13.95  | 0.58      | 0.99   |
> | TeaserGen-LR  | 20.3    | 48.69    | 0.62      | 0.87   |
>
> **[Weakness 8 & Question 11]**:
>
> The number of trainable parameters of TeaserGen-LR without the diffusion-prior model is 20.4M, where TeaserGen-LR with the diffusion-prior model has 549.8M trainable parameters. All experiments were done with one single NVIDIA RTX A6000 GPU.
>
> **[Weakness 8 & Question 12]**:
>
> We would like to point out that there are on average 2.84 scene changes per sentence, **i.e., scene changes usually happen within sentences**. Thus, our proposed method relies on textual narration in ensuring the overall coherence of the narrative, where the visuals are selected to accompany the LLM-generated textual narration. We do not want that high scene change rate as it will lead to fragmented, unengaging visuals.
> As shown in Table 4 and Table 5 (We attached the table below), we conduct **subjective evaluation** on both textual narrations and generated teasers. The subjective test shows that our generated teaser narrations are coherent, informative, and engaging, and that teasers generated by TeaserGen are overall coherent.
>
> Table 4:
>
> | Model        | Query     | Decoding    | Coherence     | Alignment     | Engagingness  | Realness      |
> |--------------|-----------|-------------|---------------|---------------|---------------|---------------|
> | TeaserGen-PT | Title     | Threshold   | 3.14$\pm$0.50     | 2.84$\pm$0.57     | 2.81$\pm$0.49     | 2.94$\pm$0.50     |
> | TeaserGen-LR | Narration | Greedy      | 2.90$\pm$0.45     | 2.88$\pm$0.48     | 2.71$\pm$0.42     | 2.71$\pm$0.44     |
> | TeaserGen-LR | Narration | Beam Search | 2.84$\pm$0.46     | 2.69$\pm$0.51     | 2.71$\pm$0.42     | 2.64$\pm$0.41     |
>
> Table 5:
>
> | Narration | Organization | Informativeness | Engagingness |
> |----------|----------|----------|----------|
> | Naive Summarization  | 3.58 $\pm$0.57  | 3.72 $\pm$0.47 | 3.60 $\pm$ 0.56|
> |Finely-tuned scripts| **3.88 $\pm$ 0.44** | **3.82 $\pm$0.54**| **3.70 $\pm$0.46**|
>
> **[Weakness 2 & Question 13]**:
>
> As detailed in Section 5.1, we introduce two independent methods, TeaserGen-PT and TeaserGen-LR, to ensure consistent matching of visual content. Specifically, TeaserGen-PT employs CLIP-ViT-B/32 to extract visual and sentence embeddings, chosen to match the embedding size of the pretrained video temporal grounding model. In contrast, TeaserGen-LR utilizes CLIP-ViT-L/14, which aligns with the embedding size required by the pretrained diffusion prior.
>
> Reference:
>
> [1] Argaw et al., Scaling up video summarization pretraining with large language models, CVPR, 2024(https://arxiv.org/abs/2404.03398)
>
> [2]  Lin et al., UniVTG: Unified Video-Language Temporal Grounding, ICCV, 2023 (https://arxiv.org/abs/2307.16715)
>
> [3] Chin-Yew Lin., ROUGE: A Package for Automatic Evaluation of Summaries, *Text Summarization Branches Out*, 2004. (https://aclanthology.org/W04-1013.)
>
> [4] Argaw et al., Towards automated movie trailer generation, CVPR, 2024(https://arxiv.org/abs/2404.03477.)

---

> ### Comment · Reviewer_VbpS · 2024-11-26
> **Response to the authors**
>
> **[Weakness 1 & Question 1]**
> The authors addressed my concern.
>
> **[Weakness 2 & Question 2]**
> Why do some metrics (e.g., REP) vary so significantly across experiments?Any thoughts/rationale?  I recommend that the authors provide a more in-depth analysis of the results, offering insights into the underlying reasons for the observed performance trends. This would enhance the clarity and impact of the findings by connecting the outcomes to the design choices and experimental setups.
>
> **[Weakness 2 & Question 3]**
>
> The authors addressed my concern.
>
> **[Weakness 4 & Question 6]**
>
> - The explanation of "20% L2 distances" is unclear. Does this mean the lowest 20% of distances or a specific percentile? The terminology requires clarification.
>
> - The rationale for selecting 15 documentaries to calculate the threshold is not explained. Why this specific number? **Is it statistically justified or arbitrary?**
>
> - Similarly, why are the top 20 images chosen based on cosine similarity? Is this number optimal, and if so, how was it determined?
>
> - **The impact of varying thresholds on model accuracy or other performance metrics is not discussed**. A quantitative description of how binary search improves the process would strengthen the argument.
>
> - How the model calculates audio-visual correspondence and the frame's importance is not described. This makes it challenging to assess the validity of the approach.
>
> - The phrase "we calculate the pixel-by-pixel L2 distance" is slightly redundant (L2 is already a element-wise operation), and its necessity after selecting semantically similar frames is unclear.
>
> - The response does not discuss the limitations or potential sources of error in the process, such as misalignment in teaser-body content matching or issues with the pretrained VTGHL model.
>
> - The response does not provide enough evidence the effectiveness of the approach. For example, **it would be helpful to include examples (qualitative) or data supporting the 0.64 VTGHLS value**.
>
> **[Weakness 5]**
>
> What was the number of beams utilized, and what impact did it have on computational overhead?
>
> **[Weakness 7 & Question 7]**
>
> My concern was addressed, I will not consider this for the final decision.
>
> **[Weakness 6 & Question 8]**
>
> - The reported ROUGE scores (ROUGE-1, ROUGE-2, and ROUGE-L) are extremely low, with values like 0.14, 0.02, and 0.12 for TeaserGen. This suggests limited overlap between the generated and reference text. However, the discussion does not critically analyze these results or explain why such low scores are acceptable.  **Why is there no significant improvement in ROUGE scores despite claims of improved narration quality?**
>
> - The narrative around the evaluators ("our carefully designed prompts") lacks transparency about the evaluation process and does not discuss possible weaknesses or biases in human evaluators. **Typically, multiple prompts are utilized, and the results are averaged to determine a statistical trend in the model's performance**.
>
> **[Question 9],  [Weakness 7 & Question 10] & [Weakness 8 & Question 11]:**
>
> The authors addressed my concerns.
>
> **[Weakness 8 & Question 12**
> - In Table 4, TeaserGen-LR using beam search scores low on coherence (2.84) and realness (2.64–2.71), suggesting a gap between textual narration coherence and its translation into a cohesive visual experience.
> - While Table 5 highlights strong subjective scores for narrations (organization, informativeness, and engagingness), these results do not directly resolve the issue of disjointed visuals caused by high scene change rates. The coherence of narration does not inherently guarantee cohesive visuals if the scene changes outpace the narration's logical flow.
>
> **[Weakness 2 & Question 13]**
> - How do TeaserGen-PT and TeaserGen-LR handle embedding mismatches or inconsistencies in practice?
> - What mechanisms are in place to update or adapt the approach as new models or embeddings become available?
>
>
> **Decision: My primary concern with this approach pertains to the 0.64 threshold, as I find insufficient evidence to support the implications of this design decision.**

---

> ### Author Response · Authors · 2024-11-27
> **Address primary concerns on 0.64**
>
> Thanks for your constructive feedback, we would like to **first address your primary concern** and then answer your questions one by one.
>
> The goal of the proposed searching algorithm is to **maximize the total video-narration correspondence while maintaining the desired length**. For **each sentence** in narration scripts, we **generate a score curve** (with the x-axis representing video time and the y-axis representing the score). This curve reflects frame importance (defined by its likelihood of being included as a highlight in video highlight detection) and text-visual correspondence to the sentence.
>
> During the search process, the algorithm selects a threshold such that the intersections of the score curve and the threshold determine the start and end times of video clips. These clips are included if their scores exceed the threshold, and multiple clips can satisfy this condition. The search starts at the highest point of the score curve, **progressively lowering the threshold in small steps (0.001) until the total duration of the selected clips matches the duration of the generated audio waveform**. This final threshold is identified as the optimal threshold of that sentence.
>
> For **each sentence** in the generated narrations, **we associate it with a score curve and determine an optimal threshold through a search process described above**. We then calculate the sentence-level VTGHLS by averaging the VTGHLS values of all frames within that sentence whose VTGHLS exceeds the threshold. Finally, we report the VTGHLS in Table 3 by averaging the sentence-level VTGHLS values across all sentences in the generated scripts of all documentaries in the test set.
>
> To improve computational efficiency, we take advantage of the monotonic increase in the total length of selected video clips as the threshold decreases. Instead of linearly decreasing the threshold, a binary search algorithm is applied to quickly identify the optimal threshold.
>
> **The estimated VTGHLS value of 0.64 for the ground truth is provided solely for comparison purposes and is not used at any stage of our training or inference pipeline**. It does **not** correspond to the optimal threshold shown in Figure 2.
>
> Below are the details of how the value 0.64 is calculated:
>
> We first match the ground truth teaser frames back to the main documentary. For each sentence in ground truth narration scripts,  we generate a score curve and average the VTGHLS of those frames within that sentence and get sentence-level VTGHLS. Then, we average the sentence-level VTGHLS values across all sentences in the narration scripts of all available ground truth teasers to obtain the estimated ground truth VTGHLS. **Since we require exact matches, we cannot rely solely on semantic meaning**. **To minimize matching errors, we experimented with several methods**:
> 1. **Pixel-by-pixel comparison**: Directly compare each frame in the ground truth teaser with all frames in the main documentary using pixel-by-pixel L2 distance.
> 2. **Semantic matching with top 10 candidates and then pixel-by-pixel comparison**: For each frame in the ground truth teaser, find the top 10 closest images in semantic meaning (i.e., the 10 images with the highest cosine similarity) and then compare those 10 images with frames in the main documentary using pixel-by-pixel L2 distance.
> 3. **Semantic matching with top 20 candidates and then pixel-by-pixel comparison**: For each frame in the ground truth teaser, find the top 20 closest images in semantic meaning (i.e., the 20 images with the highest cosine similarity) and then compare those 20 images with frames in the main documentary using pixel-by-pixel L2 distance.
>
> We found that semantic matching with the top 20 candidates resulted in the least matching error. This was verified by **manually checking the matched frames in the main documentaries against the frames in the ground truth teasers that we aimed to search for**.

---

> ### Author Response · Authors · 2024-11-27
> **Other concerns**
>
> **[Weakness 2 & Question 2]**
>
> We would like to point out that we discuss how our method addresses the issues of repetitiveness and fragmentation in Section 5.4 and in the ablation study. Here we provide two concrete examples below.
>
> We define REP as $1 - \frac{\text{Number of Unique Frames}}{\text{Total Number of Frames}}$. For example, CLIP-NN has 92.73 % repetitiveness. This is because, for each sentence, CLIP-NN generates a sentence embedding and solely relies on it to select the number of frames that match the desired length. As a result, it frequently selects the same frames, leading to high repetitiveness. In contrast, TeaserGen-PT accounts for previously selected intervals when choosing video clips for the current sentence, reducing repetition. TeaserGen-LR goes further by leveraging a diffusion prior and smoothing mechanisms, both of which are shown to effectively mitigate repetitiveness in our ablation study.
>
> We define scene change rate as $1 - \frac{\text{Number of Consecutive Frames Within the Same Scene}}{\text{Number of Frames}}$. For example, UniVTG[2] has a high scene change rate due to two limitations. First, UniVTG[2] can only process video frames of less than 10 minutes, requiring us to divide long documentaries into several short chunks and recombine them later. Second, for each chunk, UniVTG[2] selects keyframes without considering temporal continuity, resulting in high fragmentation. In contrast, TeaserGen-PT focuses on selecting important intervals, which helps reduce fragmentation. Additionally, TeaserGen-LR leverages beam search to incorporate temporal information, further improving scene continuity. By addressing both repetitiveness and fragmentation, our methods demonstrate clear advantages over baseline approaches.
>
> **[Weakness 5]**:
>
> As we shown in Section 4.2.2, we use a beam size of 5. All of our experiments were conducted on a single A6000 GPU.
>
> **[Weakness 6 & Question 8]**:
>
> 1) We would like to clarify that this is a generative task, and the reference text—the audio transcription of the ground truth teaser—is a mixture of dialogue and narration. ROUGE scores measure the overlap between our generated narrations and the ground truth teaser audio transcription. **However, our generated narrations intentionally exclude dialogue and include an engaging ending question**, which inherently reduces overlap and leads to lower ROUGE scores. **To address this limitation, we also conduct subjective evaluation metrics to better assess the quality of our generated narrations, as ROUGE alone does not fully capture the nuances of narration quality in this context**.
>
> 2) To ensure a thorough and unbiased evaluation process, we took the following steps:
>
> We compared naive summarization and finely-tuned scripts using specific GPT prompts. For naive summarization, we used the system prompt, "You are a narrator for this story." Since GPT cannot process an entire long documentary at once, we divided it into 10 chunks. For each chunk, GPT was prompted with the instruction: "Summarize the paragraph in one sentence."
> For the finely-tuned scripts, we used the same system prompt, "You are a narrator for this story," and divided the documentary into 10 chunks as well. For each chunk, we started with the same summarization step as naive summarization, instructing GPT to "Summarize the paragraph in one sentence." After generating these 10 summary sentences, we combined them into a single summarized paragraph. This combined paragraph was then further refined using an additional prompt: "Rewrite the paragraph into an engaging story opening in 10 sentences or less, keeping all names and avoiding being replaced by pronouns." Finally, we added another step to enhance the narrative by asking GPT: "Given the title {video_title} and the provided summary, formulate one thought-provoking and concise question that relates directly to the summary."
>
> **As shown in section 5.5, to assess the performance, we created two versions (A and B), each containing 5 pairs of results generated using the prompts for naive summarization and finely-tuned scripts.** We recruited 20 participants, with 11 evaluating version A and 9 evaluating version B. The results were presented in a randomized order to ensure participants were unaware of which prompt generated each result, minimizing bias. **The results of the subjective evaluations are presented as the average score, accompanied by the corresponding confidence interval to ensure a comprehensive and statistically robust evaluation**.

---

> ### Author Response · Authors · 2024-11-28
> **Continued**
>
> **[Weakness 8 & Question 12]**:
>
> We propose several modules to address the issue of disjointed visuals, and these modules have proven effective, significantly outperforming the baseline models by a wide margin.
>
> **[Weakness 2 & Question 13]**:
>
> TeaserGen-PT and TeaserGen-LR present two general paradigms for solving teaser generation tasks. TeaserGen-PT employs an interval-based approach with video-grounding backbone that can be updated. In contrast, TeaserGen-LR uses a learning-based approach that learns direct mapping across modalities.

---

> ### Author Response · Authors · 2024-12-02
>
> We sincerely thank the reviewer for the thoughtful and constructive feedback. We have carefully reviewed your comments and addressed your concerns and questions in our responses above. Additionally, we have incorporated the suggested revisions into the updated PDF. If there are any further questions or concerns, we would be happy to address them.

---

### Official Review · Reviewer_339o · 2024-11-03

**Soundness:** 2
**Presentation:** 3
**Contribution:** 2
**Rating:** 6
**Confidence:** 3

**Summary:**

The paper addresses the challenge of creating effective teasers for long videos, the authors introduce DocumentaryNet, a dataset comprising 1,269 documentaries paired with their teasers. The proposed system operates in two stages:

Teaser Narration Generation: A pretrained large language model (LLM), is prompted to create engaging, story-like narratives with a thought-provoking ending question from the transcription.
Visual Content Selection: For relevant video segments to be selected to match the narration using two methods: a pretrained contrastive language-vision model (TeaserGen-PT) and a deep learning-based sequential model (TeaserGen-LR) that aligns video frames to narration.
TeaserGen-LR frames the narration-video matching as a sequence-to-sequence learning task, where it learns a direct mapping between the sequence of sentences in the teaser narration and the video frames. It uses transformer-based architecture with a diffusion prior to embed narration and visual sequences into a shared embedding space, enabling more nuanced and context-aware matching.

The authors used both objective metrics (like F1 score, CLIPScore, and scene change rate) and subjective user surveys that evaluate in terms of consistency, informativeness, and engagingness, to assess the quality of the teasers. They found that TeaserGen-PT with threshold-based selection often provided better coherence and alignment between video and narration, while TeaserGen-LR benefited from enhanced narration-video correspondence.

**Strengths:**

1. The paper proposes frameworks to generate teasers from documentary using audiovisual alignments and scene-changes.
2. The paper demonstrates robust experiments and comparisons to baseline models.
3. The authors use thorough evaluation on their dataset using both objective metrics (like F1 score and scene change rate) and subjective evaluations (coherence, engagingness) to validate their results.
4. The paper has shown extensive ablation studies.

**Weaknesses:**

1. Limited dataset scale, both for training and testing. Though the dataset is domain specific, still the scale is limited with just 1.2k documentaries.
2. Reliance on pretrained LLM, for teaser narration generation without any check for hallucinations or error compounding due to this step.
3. The work is very domain specific, the framework’s reliance on pretrained language-vision models for narration-video alignment, while effective for documentaries, may struggle with complex visual elements that don’t directly correspond to narration, such as scenes with symbolic or artistic visuals or videos with limited narrative.
4. Comparison with other existing video summarization models is not shown in the paper.

**Questions:**

1. The paper claims generalizability of proposed models, however the work is only shown qualitatively through some examples. It would be interesting to get some quantitative results for the same.

2. Can we see the framework's results on how it adapts to other videos with complex visual elements that don’t directly correspond to narration, such as scenes with symbolic or artistic visuals or videos with limited narrative.

3. Comparison with other existing video summarization models, on how the framework is against other models on this task.

---

> ### Author Response · Authors · 2024-11-20
>
> We thank the reviewer for the thoughtful review.  We answer the reviewer’s comments below.
>
> **[Weakness 1]**
>
> Although there are only 1.2K documentaries, each documentary is over 30 minutes long, resulting in 600 hours of content. We compare our dataset with MovieNet[1] and TGT[2] in table below, we can see our dataset is **the first and largest public available teaser/trailer-body contents dataset**.
>
> | Dataset                                       | Number of Samples | Total Length (hours) | Publicly Available |
> |-----------------------------------------------|-------------------|---------------------|--------------------|
> | MovieNet[1]| 1100              | 2126.7              | No                 |
> | TGT [2]   | 22604             | N/A                 | No                 |
> | Ours                                          | 1269              | 655.7               | Yes                |
>
> **[Weakness 2]**
>
> We conduct **a subjective evaluation** of our generated narration in Section 5.5 by assessing the organization, informativeness and engagingness on a Likert scale from 1 to 5. Our carefully designed prompts are considered consistent, informative, and engaging by human evaluators. As mentioned in section 2, teaser generation is unlike traditional summarization tasks, teaser generation requires handling longer input sequences, higher compression rates, and ensuring the narration remains engaging and story-driven.
>
> | Narration            | Organization | Informativeness | Engagingness |
> |----------------------|--------------|-----------------|--------------|
> | Naive Summarization  | 3.58$\pm$0.57    | 3.72$\pm$0.47       | 3.60$\pm$0.56    |
> | Finely-tuned scripts | 3.88$\pm$0.44    | 3.82$\pm$0.54       | 3.70$\pm$0.46    |
>
> We report **common text summarization evaluation metrics**, specifically the average F1 score of ROUGE[3], in the table below. We compare our LLM-generated narration with the audio transcription of the ground truth teaser, which includes dialogues. Additionally, for reference, we compare the performance of TeaserGen narration with fully extractive narration from the main content and a naïve summarization approach.
>
> | Evaluation | Naive Summarization | Fully Extraction | TeaserGen |
> |----------|----------|----------|----------|
> | ROUGE-1 | 0.14  | 0.15 | 0.14|
> | ROUGE-2 | 0.02 | 0.02 | 0.02|
> | ROUGE-l | 0.12 | 0.14 | 0.12|
>
> **[Weakness 3 & Question 2 ]**
>
> We would like to argue that our method has broad applications in content where narration is the primary medium for conveying information, with visuals serving as a complementary and engaging element. This approach is highly applicable to documentaries, TED Talks, and educational content such as lectures in high impact application fields.
> In the table below, we can see that TeaserGen performs well in zero-shot scenarios. **We also compare TeaserGen-PT and TeaserGen-LR with two extra video summarization models: CSTA[6] and A2Summ[8]**. Note that CSTA[6] cannot process videos with more than 5,000 frames, so we chunked the long videos and recombined them after processing. Similarly, A2Summ[8] cannot handle the long documentaries in our dataset, so we divided them into 10 chunks and recombined them afterward. Since A2Summ[8] is an extractive method and ground truth is unavailable in zero-shot scenarios, we leveraged the average teaser length of approximately 1.3 minutes. For each chunk, we selected 8 key frames (fps=1) and 1 key sentence to summarize the content effectively.  We observe that TeaserGen achieves a relatively higher F1 score and also provides narrations with higher audio-visual correspondence that are effective in engaging the audience. **The generalizability of our proposed models can be further demonstrated by samples on our demo page(https://54321anonymous.github.io/ICLR2025/)**.
>
> | Model         | REP (%) | SCR (%) | CLIPScore | VTGHLS |
> |---------------|---------|---------|-----------|--------|
> | CSTA          | 4.9     | 5.4     | N/A       | N/A    |
> | A2Summ | 0 | 96.25 | 0.54 | 0.57 |
> | TeaserGen-PT  | 0       | 13.95 | 0.58      | 0.99   |
> | TeaserGen-LR  | 20.3    | 48.69   | 0.62      | 0.87   |

---

> ### Author Response · Authors · 2024-11-20
> **Continued**
>
> **[Weakness 4 & Question 3]**
>
> As shown in Table 3 (see table below), we have compared our method to a **query-based summarization model** CLIP-IT [4]. Additionally, as discussed in the paper, the recent state-of-the-art trailer generation model TGT[2] and the state-of-the-art multimodal video summarization model LFVS[5], along with their code, have not made their training datasets open-source, making direct comparison impossible. **To demonstrate the effectiveness of our proposed model, we further evaluate it against two extra SOTA video summarization model: CSTA[6] and A2Summ[8]. (Note that CSTA[6] cannot process videos with more than 5,000 frames, so we chunked the long videos and recombined them after processing. Similarly, A2Summ[8] cannot handle the long documentaries in our dataset, so we divided them into 10 chunks and recombined them afterward.)** We observe that TeaserGen achieves a higher F1 score than CSTA[6], UniVTG[7], and CLIP-IT[4] and A2Summ[8]. In addition, TeaserGen demonstrates a scene change rate more closely aligned with the ground truth and TeaserGen-LR has the highest CLIPScore.
>
> | Model         | F1(%)    | REP(%)    | SCR (%)    | CLIPScore | VTGHLS |
> |---------------|-------|--------|--------|-----------|--------|
> | CSTA        | 1.78  | 0      | 5      | N/A       | N/A    |
> | UniVTG        | 1.82  | 0      | 89.68  | 0.58      | 1.01   |
> | CLIP-IT   | 1.24  | 0      | 99.39  | 0.56      | 0.61   |
> | A2Summ | 0.77 | 0| 95.23 | 0.54 | 0.61|
> | TeaserGen-PT  | 1.85  | 0      | 13.16  | 0.56      | **1.02**   |
> | TeaserGen-LR  | **1.88**  | 19.39  | 46.56  | **0.63**      | 0.77   |
> | Ground Truth  | 100   | >7.36  | 27.6   | 0.58      | 0.64   |
>
> [1] Huang et al., MovieNet: A Holistic Dataset for Movie Understanding, ECCV, 2020. https://arxiv.org/abs/2007.10937.
>
> [2] Argaw et al., Towards automated movie trailer generation, CVPR, 2024 (https://arxiv.org/abs/2404.03477.)
>
> [3] Lin, ROUGE: A Package for Automatic Evaluation of Summaries, *Text Summarization Branches Out*, 2004. (https://aclanthology.org/W04-1013.)
>
> [4]Narasimhan et al., Clip-it! language-guided video summarization, CoRR, 2021 (https://arxiv.org/abs/2107.00650).
>
> [5] Argaw et al., Scaling up video summarization pretraining with large language models, CVPR, 2024 (https://arxiv.org/abs/2404.03398)
>
> [6] Son et al., CSTA: Spatiotemporal Attention for Video Summarization, CVPR, 2024(https://arxiv.org/abs/2405.11905)
>
> [7] Lin et al., UniVTG: Unified Video-Language Temporal Grounding, ICCV, 2023 (https://arxiv.org/abs/2307.16715)
>
> [8] He et al., Align and Attend: Multimodal Summarization with Dual Contrastive Losses, CVPR, 2023 (https://arxiv.org/abs/2303.14150)

---

### Official Review · Reviewer_uVty · 2024-11-04

**Soundness:** 2
**Presentation:** 3
**Contribution:** 3
**Rating:** 6
**Confidence:** 3

**Summary:**

This paper introduces TeaserGen, a method for generating teasers for long videos. To address the lack of suitable datasets, it presents the DocumentaryNet dataset, which contains 1,269 documentaries paired with their teasers. The dataset includes streams for video, speech, music, sound effects, and narrations. The proposed method is a two-stage system: first, it generates the teaser narration using a large language model (LLM), and then it uses vision-language models to select the most relevant visual content.

**Strengths:**

- The newly introduced dataset could be valuable for the research community.
- The task is interesting and meaningful.
- The experiments are thorough, and the performance appears good.

**Weaknesses:**

- The system heavily relies on LLMs and vision-language models, which may lead to error accumulation. How can we evaluate whether the teaser narration generated by GPT is effective?
- Some video summarization and highlight detection methods could also be applied to generate teasers, but the paper lacks a comparison with these approaches.

**Questions:**

Please refer to the weaknesses.

---

> ### Author Response · Authors · 2024-11-20
>
> We thank the reviewer for the thoughtful review. We believe our tasks are novel and have wide applications. We are encouraged by the reviewer’s positive feedback on our model performance. We address the reviewer’s comments below.
>
> **[Weakness 1]**
>
> We conduct **a subjective evaluation** of our generated narration in Section 5.5 by assessing the organization, informativeness and engagingness on a Likert scale from 1 to 5. Our carefully designed prompts are considered consistent, informative, and engaging by human evaluators. As mentioned in section 2, teaser generation is unlike traditional summarization tasks, teaser generation requires handling longer input sequences, higher compression rates, and ensuring the narration remains engaging and story-driven.
>
> | Narration | Organization | Informativeness | Engagingness |
> |----------|----------|----------|----------|
> | Naive Summarization  | 3.58 $\pm$0.57  | 3.72 $\pm$0.47 | 3.60 $\pm$ 0.56|
> |Finely-tuned scripts| **3.88 $\pm$ 0.44** | **3.82 $\pm$0.54**| **3.70 $\pm$0.46**|
>
> We report **common text summarization evaluation metrics**, specifically the average F1 score of ROUGE[6], in the table below. We compare our LLM-generated narration with the audio transcription of the ground truth teaser, which includes dialogues. Additionally, for reference, we compare the performance of TeaserGen narration with fully extractive narration from the main content and a naïve summarization approach.
>
> | Evaluation | Naive Summarization | Fully Extraction | TeaserGen |
> |----------|----------|----------|----------|
> | ROUGE-1 | 0.14  | 0.15 | 0.14|
> | ROUGE-2 | 0.02 | 0.02 | 0.02|
> | ROUGE-l | 0.12 | 0.14 | 0.12|
>
> **[Weakness 2]**
>
> As shown in Table 3 (see table below), we have compared our method to a query-based summarization model CLIP-IT [3]. We also **conducted an experiment using CLIPScore to denote the highlight score** (TeaserGen-PT-CLIP). Additionally, as discussed in the paper, the recent state-of-the-art trailer generation model TGT[4] and the state-of-the-art multimodal video summarization model LFVS[5], along with their code, have not made their training datasets open-source, making direct comparison impossible. **To demonstrate the effectiveness of our proposed model, we further evaluate it against two extra SOTA video summarization models: CSTA[1] and A2Summ[7].** (Note that CSTA[1] cannot process videos with more than 5,000 frames, so we chunked the long videos and recombined them after processing. Similarly, A2Summ[7] cannot handle the long documentaries in our dataset, so we divided them into 10 chunks and recombined them afterward.) We observe that TeaserGen achieves a higher F1 score than CSTA[1], UniVTG[2], and CLIP-IT[3] and A2Summ[7]. In addition, TeaserGen demonstrates a scene change rate more closely aligned with the ground truth and TeaserGen-LR has the highest CLIPScore.
>
> | Model         | F1(%)    | REP(%)    | SCR (%)    | CLIPScore | VTGHLS |
> |---------------|-------|--------|--------|-----------|--------|
> | CSTA        | 1.78  | 0      | 5      | N/A       | N/A    |
> | UniVTG       | 1.82  | 0      | 89.68  | 0.58      | 1.01   |
> | CLIP-IT   | 1.24  | 0      | 99.39  | 0.56      | 0.61   |
> | A2Summ|0.77|0|95.23|0.54|0.61|
> |TeaserGen-PT-CLIP| 1.31|27.23|24.10|0.58|0.74|
> | TeaserGen-PT  | 1.85  | 0      | 13.16  | 0.56      | **1.02**   |
> | TeaserGen-LR  | **1.88**  | 19.39  | 46.56  | **0.63**      | 0.77   |
> | Ground Truth  | 100   | >7.36  | 27.6   | 0.58      | 0.64   |
>
> [1] Son et al., CSTA: Spatiotemporal Attention for Video Summarization, CVPR, 2024 (https://arxiv.org/abs/2405.11905)
>
> [2] Lin et al., UniVTG: Unified Video-Language Temporal Grounding, ICCV, 2023 (https://arxiv.org/abs/2307.16715)
>
> [3] Narasimhan et al., Clip-it! language-guided video summarization, CoRR, 2021(https://arxiv.org/abs/2107.00650).
>
> [4] Argaw et al., Towards automated movie trailer generation, CVPR, 2024 (https://arxiv.org/abs/2404.03477.)
>
> [5] Argaw et al., Scaling up video summarization pretraining with large language models, CVPR 2024(https://arxiv.org/abs/2404.03398)
>
> [6] Chin-Yew Lin, ROUGE: A Package for Automatic Evaluation of Summaries, *Text Summarization Branches Out*, 2004. (https://aclanthology.org/W04-1013.)
>
> [7] He et.al., Align and Attend: Multimodal Summarization with Dual Contrastive Losses, CVPR, 2023 (https://arxiv.org/abs/2303.14150)

---

> ### Author Response · Authors · 2024-12-02
>
> We sincerely thank the reviewer for the thoughtful and constructive feedback. We have carefully reviewed your comments and addressed your concerns and questions in our responses above. Additionally, we have incorporated the suggested revisions into the updated PDF. If there are any further questions or concerns, we would be happy to address them.

---

### Author Response · Authors · 2024-11-28

We express our gratitude to the reviewers for their constructive feedback. We have made several changes to the paper (highlighted in blue) to address reviewers' concerns. Below is a summary of the strengths highlighted by the reviewers and the concerns raised by the reviewers that we have addressed in the rebuttal.

[**Strengths highlighted by the reviewers**]

1. **Innovative Approach:** A reviewer appreciates our **unique problem-solving** in automated teaser generation through TeaserGen, highlighting its creative and narration-centered two-stage approach.

2. **Comprehensive Evaluation:** Multiple reviewers commend our thorough evaluation framework, which encompasses objective metrics such as F1 score, CLIPScore, scene change rate, and repetitiveness, along with subjective assessments of coherence, alignment, engagement, and realism. This comprehensive approach effectively demonstrates the model's performance and its advantages over baseline models.

3. **Valuable Dataset:** The introduction of the DocumentaryNet dataset is noted as **addressing a critical gap in multimodal research**. This dataset, comprising 1,269 documentary-teaser pairs, serves as a valuable resource for the research community and supports advancements in teaser generation.

4. **Potential Impact:** **Reviewers highlight the potential applications of our work in media, advertising, and education**. Our approach to automated content promotion could transform how documentary content is marketed, offering practical benefits to the multimedia and educational sectors.

5. **Foundation for Future Research:** One reviewer notes that our work **lays a solid empirical foundation likely to inspire new research directions**, particularly in **long-form video modeling and advanced multimodal interaction techniques**.

[**Concerns raised by the reviewers that we have addressed in the rebuttal**]

We have carefully considered the main concerns raised by the reviewers regarding the quality of the generated narrations, the robustness of our proposed method, and the comparison with existing video summarization models. **In response, we have made several updates to our manuscript, with changes highlighted in blue for clarity**. Below is a brief summary of the modifications:

**1. Main Text:**

We clarify the research scope of this paper where we focus on narration and visual generation in this paper as an initial step toward the ultimate goal of generating teasers with visually-aligned sound effects and background music.

**2. Appendix:**

(a) **More Evaluation**:

Narration Scripts: Although we conduct subjective tests to evaluate the organization, informativeness and engagingness of our generated narration, we add an objective evaluation on LLM-Generationed  narration with ROUGE.

Zero-shot: Objective evaluation on zero-shot scenario performance to show the robustness of our proposed method. Results show that our model is more robust than existing video summarization methods in creating teasers. This can be further showcased on our demo page (https://54321anonymous.github.io/ICLR2025/).

(b) **Comparison with another two SOTA video summarization models:**

CSTA and A2Summ: We observe that TeaserGen achieves a higher F1 score than CSTA, UniVTG, and CLIP-IT and A2Summ. In addition, TeaserGen demonstrates a scene change rate more closely aligned with the ground truth and TeaserGen-LR has the highest CLIPScore.

(c) **Clarification Regarding Reviewer Feedback on the TeaserGen-PT Searching Algorithm:**

We would like point out that the estimated VTGHLS value of 0.64 for the ground truth, as reported in Table 3, is included solely for comparison purposes. It is not used at any stage in our training or inference pipeline.

(d) **More ablation study:**

Experiments on Minimum Clip length and overlapping tolerance of TeaserGen-PT

(e) **More details:**

Dataset Split Details & Comparison with Existing Teaser/Trailer-Body Contents Dataset

3. References:

CSTA: Son et al., CSTA: Spatiotemporal Attention for Video Summarization, CVPR, 2024 (https://arxiv.org/abs/2405.11905)

---

### Meta-Review · Area_Chair_5Fyo · 2024-12-23

**Metareview:**

The paper introduces a novel method for generating teaser clips, accompanied by a new benchmark. It received mixed reviews: two negative and two positive. The concerns raised were mostly minor, focusing on engineering aspects and stemming from misunderstandings. Although the authors' rebuttal appears to have effectively addressed these issues, the reviewers have not actively engaged with the rebuttal. Given that the authors successfully resolved the concerns, the AC concludes that the merits of the proposed work outweigh its flaws.

**Additional Comments On Reviewer Discussion:**

One of the reviewers assigned a score of 3, expressing several concerns which were mostly addressed in the initial author rebuttal. Although the reviewer acknowledged that many issues were resolved, they continued to express concern over the manual selection of a threshold set at 0.64, questioning its justification. The authors clarified in a subsequent response that this threshold was not used during training or inference, but solely for comparison purposes to aid reader comprehension of the results. Unfortunately, there was no further response from the reviewer.

The AC believes that the authors' responses adequately addressed the concerns raised, and attributes the persistently low score to the reviewer's lack of further engagement.

---

### Decision · Program_Chairs · 2025-01-22

Accept (Poster)